# Deficient humoral responses and disrupted B-cell immunity are associated with fatal SFTSV infection

Peixin Song[1], Nan Zheng[2,3,4], Yong Liu[5], Chen Tian[1], Xilin Wu[2,4], Xiaohua Ma[6], Deyan Chen[2,3], Xue Zou[2,3], Guiyang Wang[1], Huanru Wang[2,3], Yongyang Zhang[1], Sufang Lu[1], Chao Wu[1] & Zhiwei Wu[2,3,4]

Severe Fever with Thrombocytopenia Syndrome (SFTS), an emerging infectious disease caused by a novel phlebovirus, is associated with high fatality. Therapeutic interventions are lacking and disease pathogenesis is yet to be fully elucidated. The anti-viral immune response has been reported, but humoral involvement in viral pathogenesis is poorly understood. Here we show defective serological responses to SFTSV is associated with disease fatality and a combination of B-cell and T-cell impairment contribute to disruption of anti-viral immunity. The serological profile in deceased patients is characterized by absence of specific IgG to viral nucleocapsid and glycoprotein due to failure of B-cell class switching. Expansion and impairment of antibody secretion is a signature of fatal SFTSV infection. Apoptosis of monocytes in the early stage of infection diminishes antigen-presentation by dendritic cells, impedes differentiation and function of T follicular helper cells, and contributes to failure of the virus-specific humoral response.

[1] Department of Infectious Diseases, Nanjing Drum Tower Hospital, Nanjing University Medical School, Nanjing, People's Republic of China. [2] Center for Public Health Research, Nanjing University Medical School, Nanjing, People's Republic of China. [3] State Key Lab of Analytical Chemistry for Life Science, Nanjing University, Nanjing, People's Republic of China. [4] Jiangsu Laboratory for Molecular Medicines, Nanjing University Medical School, Nanjing, People's Republic of China. [5] Department of Experimental Medicine, Nanjing Drum Tower Hospital, Nanjing University Medical School, Nanjing, People's Republic of China. [6] Y-Clone BioMedical, Ltd., Suzhou Hi-Tech Innovation Park, Suzhou, People's Republic of China. These authors contributed equally: Peixin Song, Nan Zheng. Correspondence and requests for materials should be addressed to Z.W. (email: wzhw@nju.edu.cn)

Severe Fever with Thrombocytopenia Syndrome (SFTS), an emerging infectious disease, is caused by a novel member of phlebovirus, Bunyaviridae family[1]. Increasing SFTSV infection has raised serious concerns in East Asia[2]. The reported fatality of SFTSV infection, ranging from 10 to 30%, is significantly higher than that of the hemorrhagic fever caused by Hantavirus in the same area[3,4]. Although the virus is mainly transmitted by tick bites, human-human transmission has been reported[5,6]. No effective therapies or vaccines are available yet, and the mechanisms of disease pathogenesis are poorly understood. Although previous studies revealed that the impairment of innate immune response as well as inflammatory cytokine storm played important roles in the disease progress[7,8], our knowledge of virus-specific humoral response to SFTSV infection and its roles in the pathogenesis is extremely limited.

B-cell-dependent immunity is regulated by antigen-presenting cells (APCs) and follicular helper T cells (Tfh)[9]. Previous studies revealed that dendritic cell (DC)-restricted antigen presentation alone could initiate the Tfh cell program but could not complete ultimate effector differentiation[10]. However, the cooperation of MHC-II-positive DCs and B cells could realize the optimal effect in Tfh and germinal center (GC) B-cell differentiation in response to viral infection[11]. Tfh, along with follicular DCs, repeatedly undergo intimate interactions with the antigen-experienced B cells in GC to facilitate the B-cell expansion, differentiation and affinity maturation of plasmablast and memory B cells for the generation of high affinity antibodies[12,13]. Therefore, Tfh, mainly residing at the GC of lymph node (LN) and spleen (SP)[13], play the pivotal role in the production of pathogen-specific class-switched neutralizing antibodies. A number of earlier studies demonstrated that peripheral Tfh (pTfh) cells exhibited transcriptional and phenotypic similarities to lymphoid Tfh cells[9,14]. We, therefore, investigated the regulatory status of Tfh in the peripheral blood of SFTS patients.

In addition to the cognate interaction among immune cells, regulatory cytokines such as IL-4, GM-CSF, IL-21 and IL-6 are also required for the generation and maintenance of neutralizing antibody responses. In a humanized mice model with immature B cells and deficient CD209+ (DC-SIGN) human DCs, expression of human GM-CSF and IL-4 could correct the defects of IgG response to antigen stimulation[15]. In addition, IL-4 and IL-21 have been shown to be the fundamental effectors of Tfh cells in Th2 humoral response[16]. IL-21 has been shown to be a critical marker of Tfh cells in both phenotype and function[9,13]. Interestingly, IL-6, a proinflammatory cytokine, is essential for escalating cell response to control a persistent viral infection[17]. Nevertheless, the roles of these cytokines in the humoral response to SFTSV infection are poorly understood.

Recent research showed that SFTSV effectively infected monocytes and interfered with signaling pathway of innate immunity, which impacted on adaptive immune response[18]. Our previous work also showed that SFTSV infection impeded the differentiation of myeloid DCs[8]. The observation implies the impairment of the professional APC. Considering the critical importance of APC in the establishment of adaptive immune response, we postulate that there are defects in the humoral response induced by SFTSV infection. In the current study of a patients' cohort, we examined the dynamic nature of serologic response, modulation of B-cell subsets, myeloid DCs (mDCs) and pTfh cells, as well as several related regulatory cytokines, to elucidate the status of B-cell dependent immune response and its roles in the pathogenesis of this virulent viral infection.

## Results

**Impaired antibody responses to Gn and NP in deceased cases.** Between April and October of 2016, 43 patients admitted into

Nanjing Drum-Tower Hospital were confirmed with SFTSV infection. The peripheral blood samples of 30 patients, of whom 11 were deceased, were collected at multiple time points ranging from 3 to 18 days post symptom onset. Longitudinal serum antibody responses to NP and Gn of SFTS virus were determined by both ELISA and western blot (WB). Interestingly, the NP-specific IgM and IgG were completely absent from the deceased group from the symptom onset to death (Fig. 1a, b). In contrast, NP-specific IgM was positive at early time points after symptom onset and remained positive during the entire hospital stay in all recovered patients. Among this group, 7 out of 10 patients had serum antibodies reaching the highest titers in the 1st week post symptom onset. As for serum NP-specific IgG (Fig. 1a, c, upper panel), although four out of ten convalescent patients were negative in the 1st week, all of them became positive from the 2nd week post symptom onset and manifested increasing IgG titers during the 3-week clinical course.

We further analyzed serum antibody to Gn and Gc, two viral structural glycoproteins that were expressed in mammalian cells and purified (Supplementary Fig. 1a and 1b). In accordance with serological response to NP, the titers of Gn-specific IgG in recovered patients manifested increasing trend during the 3-week period after symptom onset, as compared with negative results in all deceased patients for the same period (Fig. 1d), which were confirmed by WB detection of Gn-IgG during the 3rd week post symptom onset (Supplementary Fig. 1c). Distinct from IgM response to NP, Gn-specific serum IgM was positive in all deceased patients and in five out of ten survived patients in the 1st week (Supplementary Fig. 1d), but became undetectable in all patients in the following two weeks. Interestingly, none of the patient sera reacted with Gc. Together, the absence of both NP- and Gn-specific IgG in the deceased patients indicates that the defective serological responses to the pathogen are associated with the fatal outcome (Figs. 1e, f).

**Deficient specific IgG contributed to persistent viremia.** To investigate if the defective antibody responses were associated with viremia, serum viral loads were determined at the corresponding time points. The quantitative RT-PCR analysis demonstrated that the serum viral loads of all survived patients gradually decreased during acute phase of SFTS, and became undetectable within 7–12 days after symptom onset (Fig. 2a). Strikingly, all deceased patients maintained continuously high levels of viremia from symptom onset until they succumbed to the disease. The viral loads of the deceased group were significantly higher than the survived group in the 3-week period post symptom onset ($p < 0.005$). The mean $\log_{10}$ values of the viral load in the deceased patients were 7.5, 8.2 and 8.0 for the 1st, 2nd and 3rd weeks, respectively, as compared with 5.3, 1.9 and 0 in the survived patients (Fig. 2a). NP-IgG titers had strong association with serum viral load as shown by correlation analysis of the dynamic data of 10 survived and 5 deceased patients (Fig. 2b), suggesting that virus-specific IgG exerted crucial effect in viral clearance. However, the correlation between NP-IgM titers with the viral load was not significant (Fig. 2b). In order to confirm the role of virus-specific antibodies in the clearance of virus, we further investigated the neutralization activity of Gn-specific antiserum from an immunized camel, and sera from a convalescent patient and a deceased patient (Fig. 2c). The results showed that the sera of both Gn-immunized camel and the convalescent patient effectively neutralized SFTSV, with 50% FRNT value at $0.118 \pm 0.001$ and $0.009 \pm 0.002$, respectively. In comparison, the serum from the deceased patient failed to show any significant neutralization.

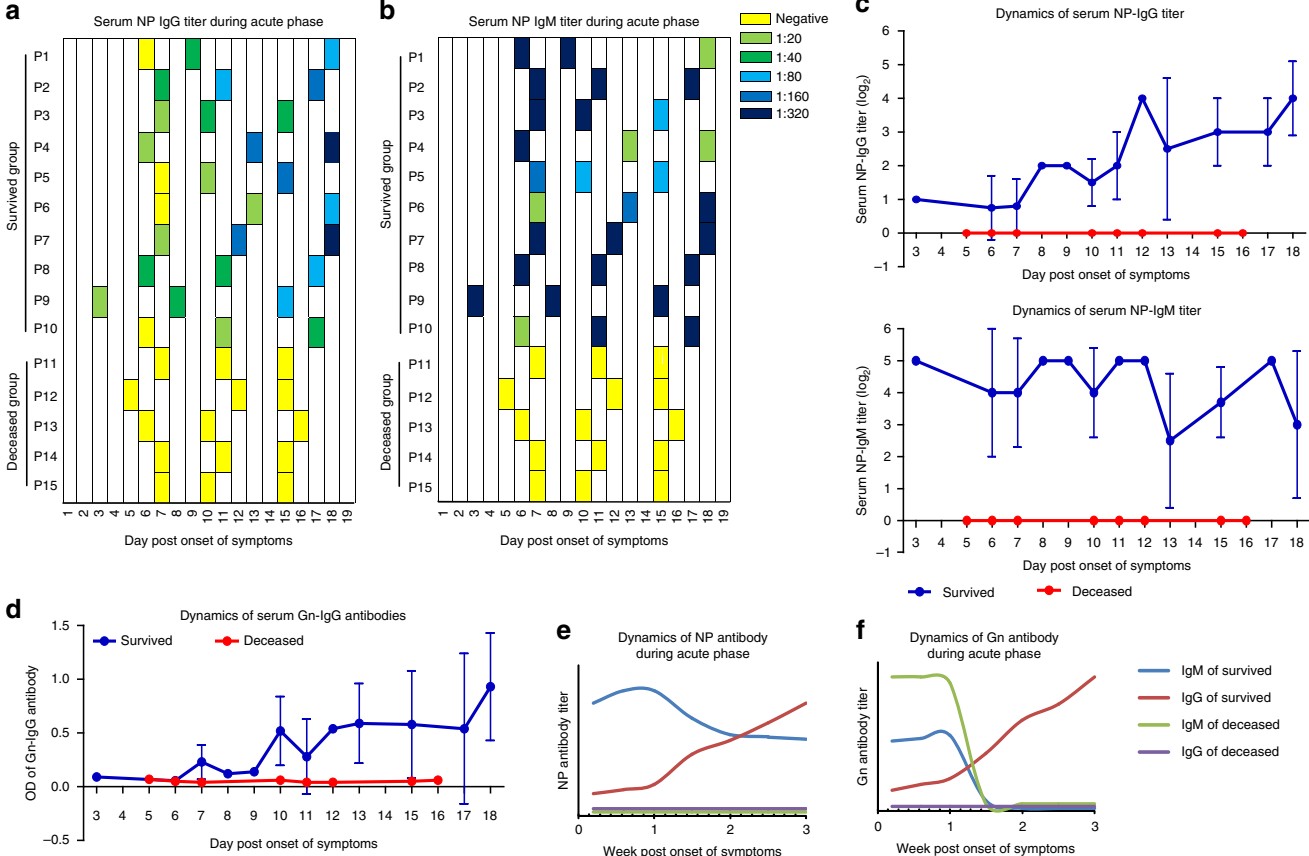

**Fig. 1** Serological response to nucleocapsid protein (NP) and glycoprotein (Gn) during acute SFTSV infection. **a**, **b** Kinetics of serum IgG (**a**) and IgM (**b**) titers to NP as determined by a commercial ELISA kit in 10 survived and 5 deceased patients from 3 to 18 day post symptom onset. **c** Clustered data on the dynamics of IgG and IgM titers to NP in 10 survived and 5 deceased patients from 3 to 18 day post symptom onset (means ± SD). **d** Dynamics of IgG titers to Gn in 10 survived and 5 deceased patients from 3 to 18 day post symptom onset (means ± SD). **e**, **f** Serum antibody titers to NP (**e**) and Gn (**f**) in SFTS patients vs. time (in week) during the acute phase, based on the average titer of NP-antibody or the positive rate of Gn-specific immunoreactive bands at different days. Error bar represents the standard deviation

**Dysregulation of peripheral B-cell subsets in fatal patients**. To understand how the peripheral B-cell subsets were impacted with respect to humoral response during acute SFTSV infection, we analyzed various B-cell subtypes in PBMCs by flow cytometry (Fig. 3a and Supplementary Fig. 2). B-cell subsets were defined phenotypically as reported[19,20]: naive (CD19+IgD+CD27−), memory (MB, CD19+IgD−CD27+), plasmablast (PB, CD19+IgD−CD27high), marginal zone-like B cells (MZ, CD19+IgD+CD27+), and double-negative B cells (CD19+IgD−CD27−). The composition of total B cells in acute phase of SFTS was presented in Fig. 3b. We and others had previously reported that disease outcome did not impact on the numbers of patient's leukocytes and lymphocytes[8,21]. The regulation of the proportion and numbers of each B-cell subset also exhibited high consistency. The proportion of naive B cells within total peripheral B cells of the deceased and the survived groups was significantly lower than that of the healthy control in the 3-week period post symptom onset (Fig. 3c). Interestingly, the mean weekly frequency of the naive B cells in the deceased group dramatically reduced to only 6.8, 5.7, and 5.2 in the 1st, 2nd and 3rd weeks, respectively, in contrast to 25.4, 23.3, and 21.6 in the survived group. This observation suggests an overwhelming activation and exhaustion of naive B cells in the process of robust viral replication.

We then evaluated the dynamics of two important effector B-cell subsets, PB, and MB cells, both of which differentiate from naive B cells (Figs. 3d, e). The frequencies and numbers of both PBs and MBs subsets in both survived and deceased patients were

significantly higher than that of the healthy control. Although the fraction of MBs in total B cells did not differ between the deceased and the survived groups, the fraction of PBs in the deceased group increased significantly more than the survived group during the entire acute phase (Fig. 3d). The significant increase of PBs is in stark contrast to the absence of serological response in the deceased patients, but nevertheless it is consistent with an earlier report that defective T-cell participation led to a surge of PBs during VSV infection[22]. The MZ B cells, another relatively small but essential subset, were significantly lower in the deceased group than that in both the survived and the healthy groups, and manifested a decline with the progression of SFTS (Fig. 3f).

The frequency of IgD−CD27− subset notably increased with the disease aggravation in the deceased group, and reached 56.7% of total B cells in the 3rd week as compared with 40.0% in the survived group (Fig. 3g). Further investigation of the IgD−CD27−B-cell compartment (Fig. 3h) demonstrated that the mean frequency of IgM+ cells in the survived, deceased and healthy control groups was 26.8, 19.6, and 31.0%, respectively, as compared with the corresponding frequency of IgG+ cells at only 4.9, 3.5, and 1.3%. However, both the IgG and IgM fractions in IgD−CD27−B cells showed no significance difference among three groups (Fig. 3i). The CD21 negative fraction of IgM+IgD−CD27−B cells in the deceased group was 89.9%, and significantly higher than in the other two groups. IgM+IgD−CD27−B cells were phenotypically defined as B-cell precursors

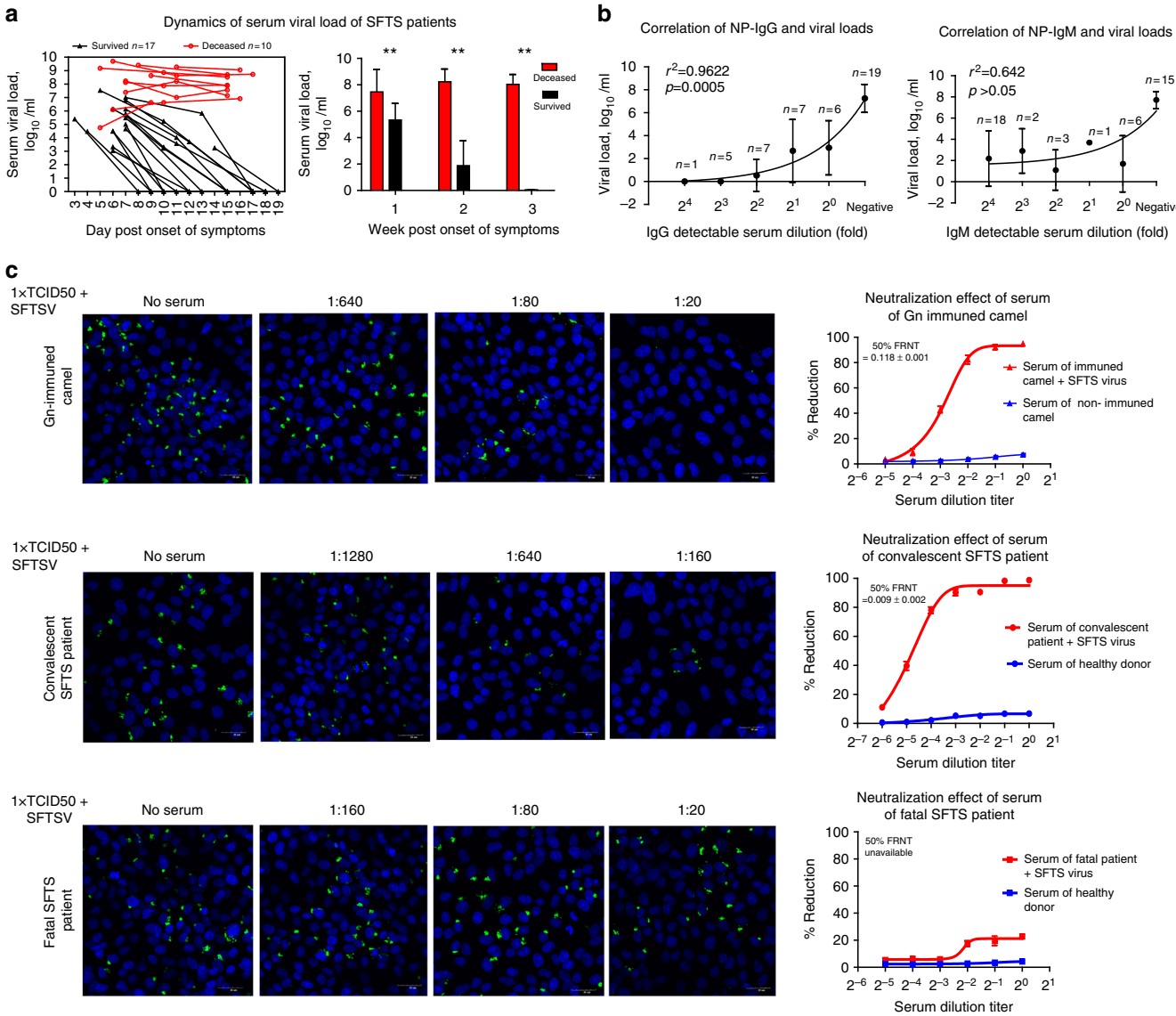

**Fig. 2** Correlation between serum viral load and virus-specific antibodies. **a** Serum viral load of 17 survived and 10 deceased patients at different day point was determined by RT-PCR, and then their kenetics of viremia during acute phase of SFTS was generated in the left panel. The level of serum viral load of the survived and deceased group during the 3-week clinical course was compared using $T$ test in the right panel, ** $p < 0.005$. **b** Linear regression analysis of correlations between viral load and NP-specific antibodies. $X$-axis denotes the dilution fold of the patients' serum samples that gives detectable signal of NP-specific antibodies, and negative means undetectable antibodies at any dilution fold. $r$ and $p$ indicate the correlation coefficient and the $p$ value of significance, respectively. $p < 0.05$ is considered as significant. **c** Neutralization of SFTSV by sera from the Gn-immunized camel, convalescent and fatal patients were shown by fluorescence staining (left panel), and percentage of infection reduction as indicated in 50% FRNT value (right panel). Non-specific neutralization effects of both human and camel sera were shown in blue. The data were from two independent experiments, and each data point from duplicate wells (means ± s.e.m). Error bar represents the standard deviation

in bone marrow[23], and the expression of CD21 in this subset could reflect the degree of maturation during B-cell ontogeny[24]. Therefore, due to the extremely large fraction of $CD21^-$ B cells in $IgM^+IgD^-CD27^-$ subset of the deceased patients, we inferred that the dysfunction of pre-B-cell maturation might be involved in the dramatic reduction of mature naive B cells with $CD27^-IgD^+$ phenotype.

**Failed antibody class-switch response in fatal patients**. To further characterize the roles of PBs and MBs in SFTSV infection, we examined immunoglobulin expression by both PB and MB subsets[25,26] by intracellular staining of IgG and IgM (Fig. 4a). PBs and MBs as percentage of total B cells were shown in

Supplementary Fig. 3. The kinetics of PBs and MBs as defined by CD38 shown in Fig. 4b were highly consistent with the result of Figs. 3d, e. PBs of the survived patients elevated significantly in the 1st and 2nd weeks post symptom onset as compared with the healthy control, and subsequently decreased to normal level in the 3rd week. The similar phenomenon was also reported in Ebola virus infection[25]. The magnitude of PB expansion in the deceased group was markedly higher than that in the survived group during all three weeks. The proportion of MBs showed no significant differences between the two groups of patients as shown in Fig. 4b.

Analysis of immunoglobulin expression by FACS demonstrated no significant differences in class-unswitched $IgM^+IgG^-$ PBs among three groups (Fig. 4b). However, class-switched IgM

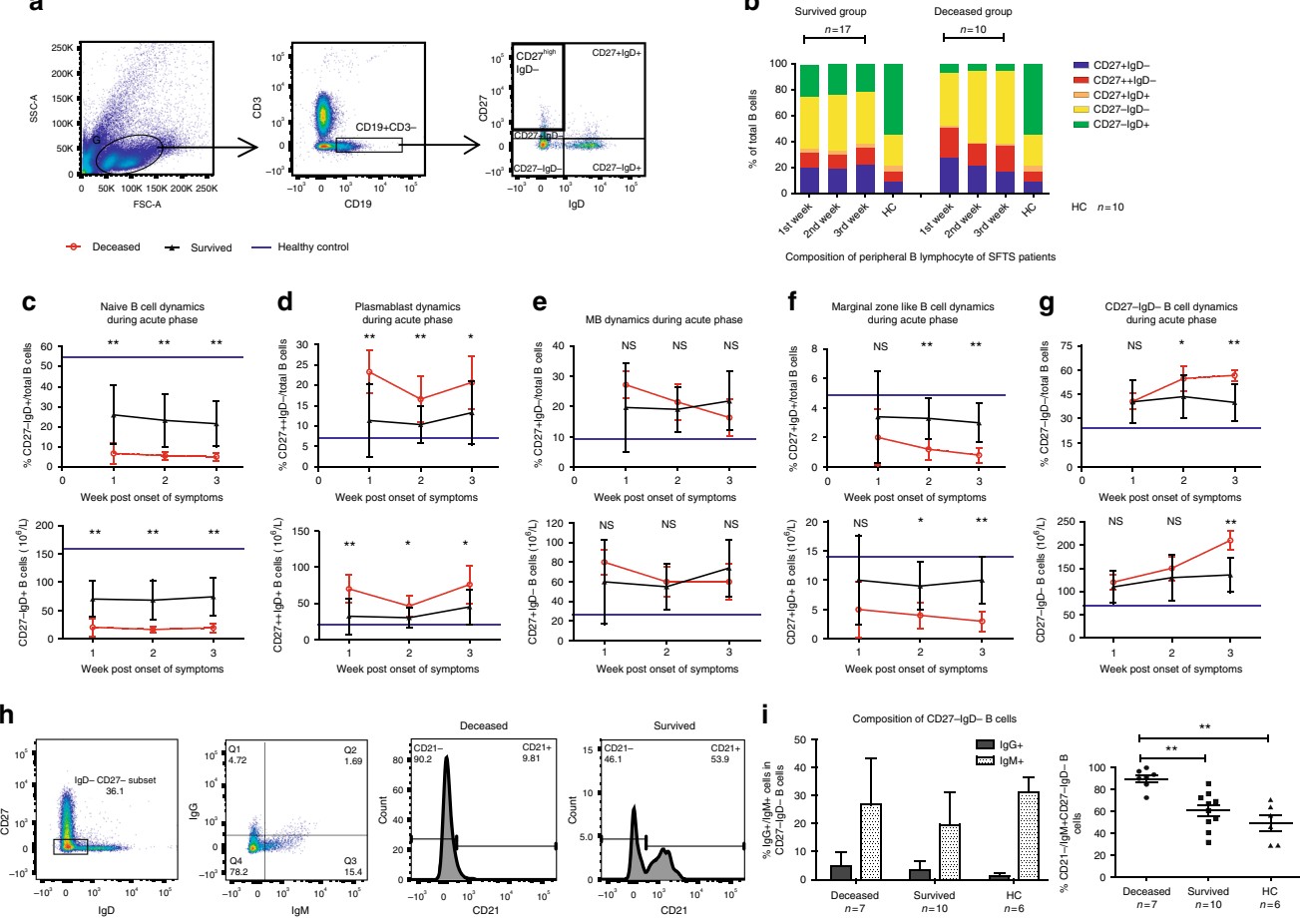

**Fig. 3** Kinetics of peripheral B-cell subsets during acute phase of SFTS. **a** FACS gating strategy to define B-cell subsets by CD27 and IgD. **b** Schematic presentation of the relative composition of B-cell subsets in total peripheral B cells in the 3-week period post symptom onset. IgD−CD27+, IgD−CD27++, IgD+CD27+, IgD−CD27− and IgD+CD27− represent memory, plasmablast, marginal zone-like, double negative and naive B-cell subsets respectively. HC = healthy control. **c**–**g** Kinetics of the proportion (upper) and numbers (lower) of B-cell subsets in total B cells and comparison of each subset between survived and deceased patients. Black triangles, red circles and blue horizontal lines represent means of survived, deceased and HC group, respectively. Statistic analysis was conducted between the survived and deceased groups. **h** FACS gating strategy defining IgG, IgM and CD21 expression in CD27−IgD− subset of B cells. **i** Expression of IgG and IgM in CD27−IgD− B cells (left) and the ratio of CD21− subset among IgM-producing CD27−IgD− B cells (right), shown as means ± SD. *$p < 0.05$, **$p < 0.005$. Error bar represents the standard deviation; NS = no significance

−IgG+ PBs in the survived patients exhibited steady elevation from the 1st week, and continued to the 2nd week, as compared with the drastic decline in the deceased patients, which was even significantly lower than the healthy control (Fig. 4b). It is interesting that a large fraction of IgM−IgG− PBs in PBMCs was observed in the deceased patients, representing 71.2, 54.6, and 65.2 of the total PBs in the 1st, 2nd, and 3rd weeks, respectively, as compared with only 31.3, 31.1, and 30.6% in the survived group (Supplementary Fig. 3g). Considering that PBs could secret other immunoglobulin isotypes besides IgG and IgM[27], we further measured IgA and IgD expression in these PBs (Fig. 4c–d) and found that only low levels of IgD+ and IgA+ fractions were detected in the PBs and they showed no significant difference between the two patient groups. Taken together, we conclude that a relatively large fraction of CD27+CD38+ B-cell subset in the deceased patients didn't secret IgG, IgM, IgD, and IgA isotypes.

The transcriptional profile of PBs was investigated by analyzing the expression of BLIMP-1, IRF-4, and XBP-1, three key genes involved in the antibody-secreting network[27]. When proceeding to the antibody-secreting phase, B cells significantly upregulate the expression of BLIMP-1, IRF-4 and XBP-1. The result showed

that the expression of all three genes in PBs was significantly down-modulated in the deceased group (Fig. 4e–f). To further illustrate the functional regulation of ASC in SFTS patients, we performed an Elispot assay, which captures virus-specific IgG after cell stimulation in vitro using samples from three patients, representing different temporal phases and clinical outcomes. R848, an TLR agonist, was used as the non-specific stimulator[28]. The samples of two survived patients, P28 and P29, were collected at the early and later phases of acute SFTSV infection, respectively. The sample of patient P30 was collected one day before death. PB of two survived patients, but not of the deceased patient, produced Gn-specific IgG (Fig. 4g). Moreover, the stimulation with the combination of R848 and Gn protein produced more specific IgG secretion than with only R848 in the survived patients.

The kinetics of IgM and IgG expression in MBs showed that in the survived patients the fraction of IgM+IgG− MBs gradually declined with IgG+IgM− MBs significantly elevated until the 3rd week post symptom onset (Fig. 4b, middle at lower panel), indicating a later class-switching response in MBs than that in PBs. By comparison with the healthy control, the proportion of

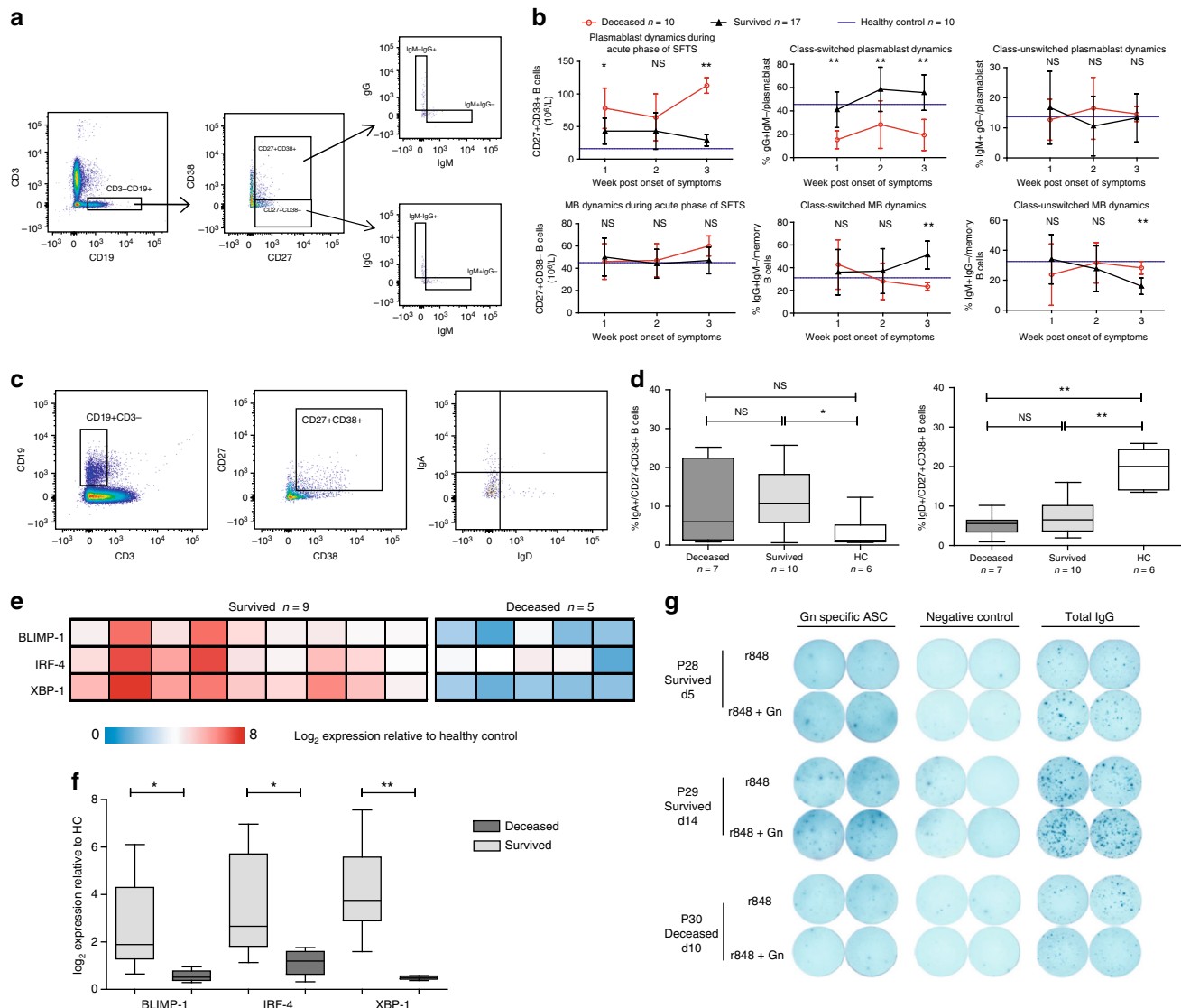

**Fig. 4** Phenotypic and functional analysis of antibody-secreting cells during acute phase of SFTS. **a** FACS gating strategy to define plasmablasts and memory B cells by CD27 and CD38, as well as immunoglobulin staining of PBs and MBs. **b** Kinetics of the numbers of PBs (upper left panel) and MBs (lower left panel), and phenotypic analysis of immunoglobulin class-switching response of ASCs of survived and deceased group. IgM$^+$IgG$^-$ and IgG$^+$IgM$^-$ subsets represent the class-unswitched (right panels) and class-switched (middle panels) ASCs, respectively. Black triangles, red circles and blue horizontal lines represent means of survived, deceased and HC group respectively. Statistic analysis was conducted between the survived and deceased groups. *$p < 0.05$, **$p < 0.005$. **c** FACS gating strategy defining IgA and IgD expression in CD27$^+$CD38$^+$ subset of B cells. **d** The ratio of IgA$^+$ or IgD$^+$ cells in CD27$^+$CD38$^+$ subset of B cells among the deceased, survived patients and healthy donors (HC), shown as means ± SD. **e**, **f** Expression levels of transcription factors BLIMP-1, IRF-4 and XBP-1 were determined in the CD27$^+$CD38$^+$ subset of B cells from patients or healthy donors (HC) as shown in heatmap (**e**) or bargraph (**f**). Statistic analysis was conducted by t test. *$p < 0.05$, **$p < 0.005$, NS = no significance. **g** ELISPOT analysis for the detection of Gn-specific ASCs of three representative patients (two survivors and one deceased). The fresh peripheral blood samples of three patients (P28, P29, and P30) were collected at day 5, 14, and 10 post symptom onset, respectively. The ELISpot analysis was performed as described in the Methods section. The survived patients (P28 and P29) showed positive Gn-specific response as compared with corresponding negative control, and the deceased patient (P30) showed negative result. Error bar represents the standard deviation

both IgM$^+$IgG$^-$and IgG$^+$IgM$^-$ MBs in the deceased patients did not show any meaningful regulation.

**Robust pTfh differentiation emerged in survived patients.** To understand the roles of Tfh in the B-cell immunity against SFTSV, we defined the pTfh phenotypically as CD3$^+$CD4$^+$ICOS$^+$CXCR5$^+$PD-1$^+$[9] and gated them on PBMCs as shown in panel D (Fig. 5a). We analyzed IL-21 expression in pTfh after in vitro stimulation with Leukocyte Activation Cocktail by FACS and RT-PCR. The number of pTfh in the survived, but not the deceased patients showed robust increase in the 1st week, as compared with

that in the healthy control (Fig. 5b, right panel), and then gradually decreased in the following two weeks to the baseline (Fig. 5b), suggesting that pTfh expansion in the early stage of acute SFTVS infection is associated with the recovery of the disease. Moreover, the expression of ICOS, a surface molecule for Tfh migration to GC[29–31], showed no significant difference among three groups (Fig. 5c, left panel). However, PD-1 expression on pTfh in the deceased patients maintained a significantly higher level than that in both the survived patients and the healthy control (Fig. 5c, right panel). The expression of ICOS, PD-1 and IL-21 by individual patients as measured by FACS was shown in Supplementary Fig. 6.

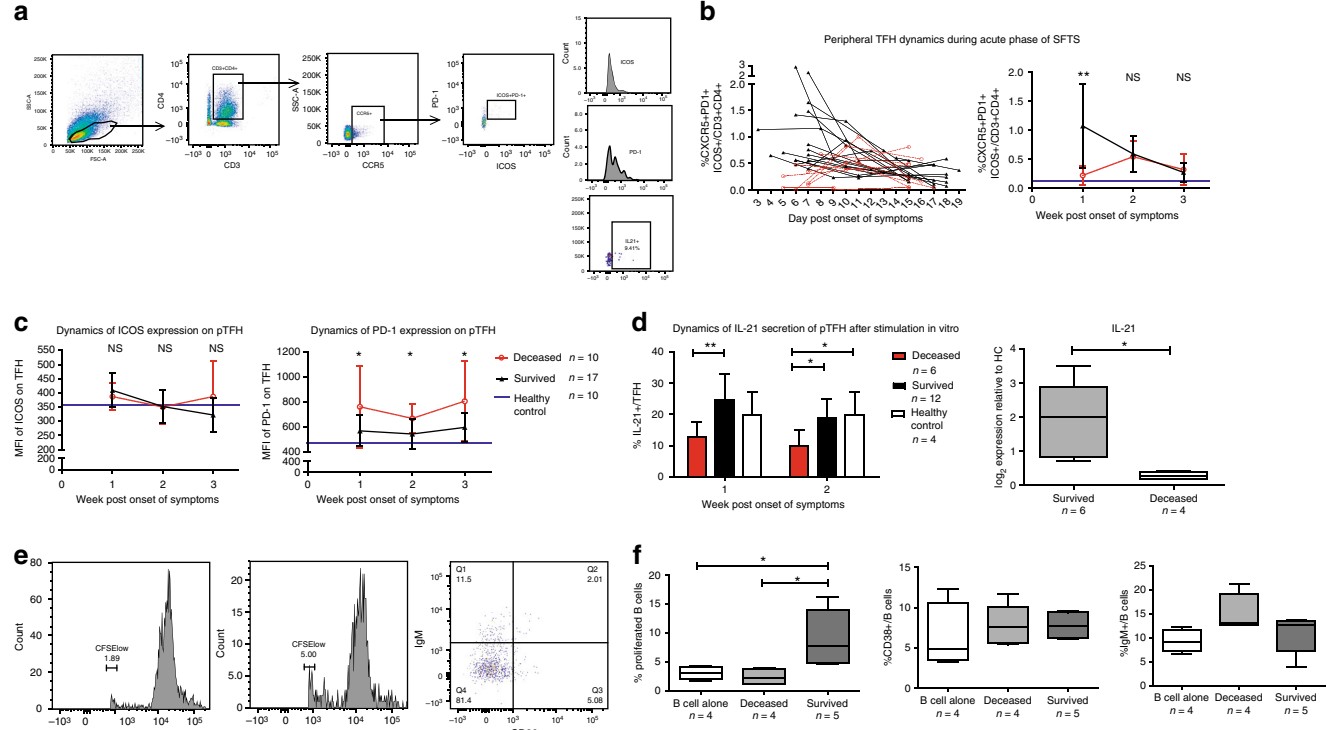

**Fig. 5** Peripheral Tfh response during acute phase of SFTS. **a** FACS gating strategy to define IL-21+ pTfh cells. Histogram illustrates the surface expression of ICOS and PD-1 on pTfh cells. **b** Kinetics of pTfh differentiation from 3 to 18 days post symptom onset and the comparison of the percentages of CXCR5+PD-1+ICOS+ in CD3+CD4+ cells for the 3-week period post symptom onset among survived (black triangle) and deceased (red circle) patients and healthy individuals (purple line). Symbols represent means of different groups. **c** Comparison of the expression of ICOS and PD-1 on pTfh cells measured as MFI among survived (black triangle) and deceased (red circle) patients and healthy individuals (purple line) during acute phase of SFTS. Statistic analysis was conducted between the survived and deceased groups. **d** IL-21 was only observed in pTfh population, and the proportion of IL-21 positive cells in total Tfh cells was compared as indicated (left). In addition, IL-21 expression in pTfh cells was determined by qPCR (right). The quantitative level of mRNA was normalized to β-actin using the cycle threshold (Ct) method ($2^{-\triangle\triangle Ct}$ method), then compared in $\log_2$ value. *$p < 0.05$, **$p < 0.005$, NS = no significance. **e**, **f** Proliferation, IgM and CD38 expression of naive B cells as activated by coculture with autologous pTfh cells as measured by FACS. **f** Proliferative cells were shown as CFSE low subsets in the CFSE- labelled naive B cells. Statistics of the proliferation and IgM/CD38 expression in the naive B cells co-cultured with pTfh cells were shown. Significant difference was calculated using one way ANOVA. *$p < 0.05$. Error bar represents the standard deviation

FACS analysis of IL-21 expression in pTfh of selected SFTS patients (6 deceased and 12 survived) demonstrated that the proportion of IL-21+ pTfh was significantly higher in the survived patients than that in the deceased patients in the 1st and 2nd weeks post symptom onset (Fig. 5d, left panel). In addition, RT-PCR analysis also showed that the relative gene expression of IL-21 in the sorted pTfh of the survived patients was significantly higher than that of the deceased patients (Fig. 5d, right panel). To further explore the influence of pTfh on B-cell maturation, we performed a CD4+ B-cell helper assay by placing the pTfh sorted from SFTS patients in co-culture with autologous naive B cells, and assessed proliferation and CD38/IgM expression by FACS (Fig. 5e). Although both CD38 and IgM expression showed no significant differences in B cells between the survived or deceased patients, the greater proliferation of naive B cells was still observed when co-cultured with autologous pTfh from the survived individuals (Fig. 5f). Collectively, these data indicated the impairment of Tfh in promoting B-cell immunity during fatal SFTSV infection.

**Severe monocyte apoptosis and inhibited mDC differentiation**. Monocytes represent the most abundant circulating pool of DCs[19], which play critical roles in the development of adaptive immune response by presenting antigens and providing co-stimulatory signals to both T and B cells[10]. We, therefore,

analyzed apoptosis of the adherent cells of PBMC, which consist of about 90% of monocytes[32,33] in the 1st week post symptom onset. Strikingly, the positive rates of both annexin V and PI in the adherent cells from the deceased patients were significantly higher than those in the survived patients and the healthy control (Fig. 6a, b), indicating monocytes undergoing severe apoptosis at an early stage of SFTSV infection. Moreover, in vitro infection assay indicated that the annexin V+PI− and annexin V+PI+ proportions of SFTSV-infected mDCs, representing the early apoptotic cells and late apoptotic cells, respectively, were significantly higher than that with mock infection (Fig. 6c, d).

We then investigated the differentiation and function of mDCs by FACS in patients' blood samples. mDCs were gated as CD14−B220−CD11c+CD123−HLA-DR+ on PBMCs and B220− was used as a lineage control (Fig. 6e) since B cells will be the major interference to the selection of CD14−CD11c+CD123−HLA-DR+ as mDCs' markers[34]. The robust differentiation of mDCs in the survived, but not in the deceased patients, initiated at the 2nd week and sustained to the 3rd week post symptom onset (Fig. 6f and Supplementary Fig. 4a). The expression of co-stimulatory molecule CD86 on mDCs from the 2nd week and the proportion of CD80+CD86+/mDCs in the 3rd week of the survived patients were significantly higher than that of the deceased patients, as the expression of CD80 showed no meaningful difference between both patients' groups (Fig. 6g, h). These data suggest that the differential activation of mDCs in the survived and the deceased

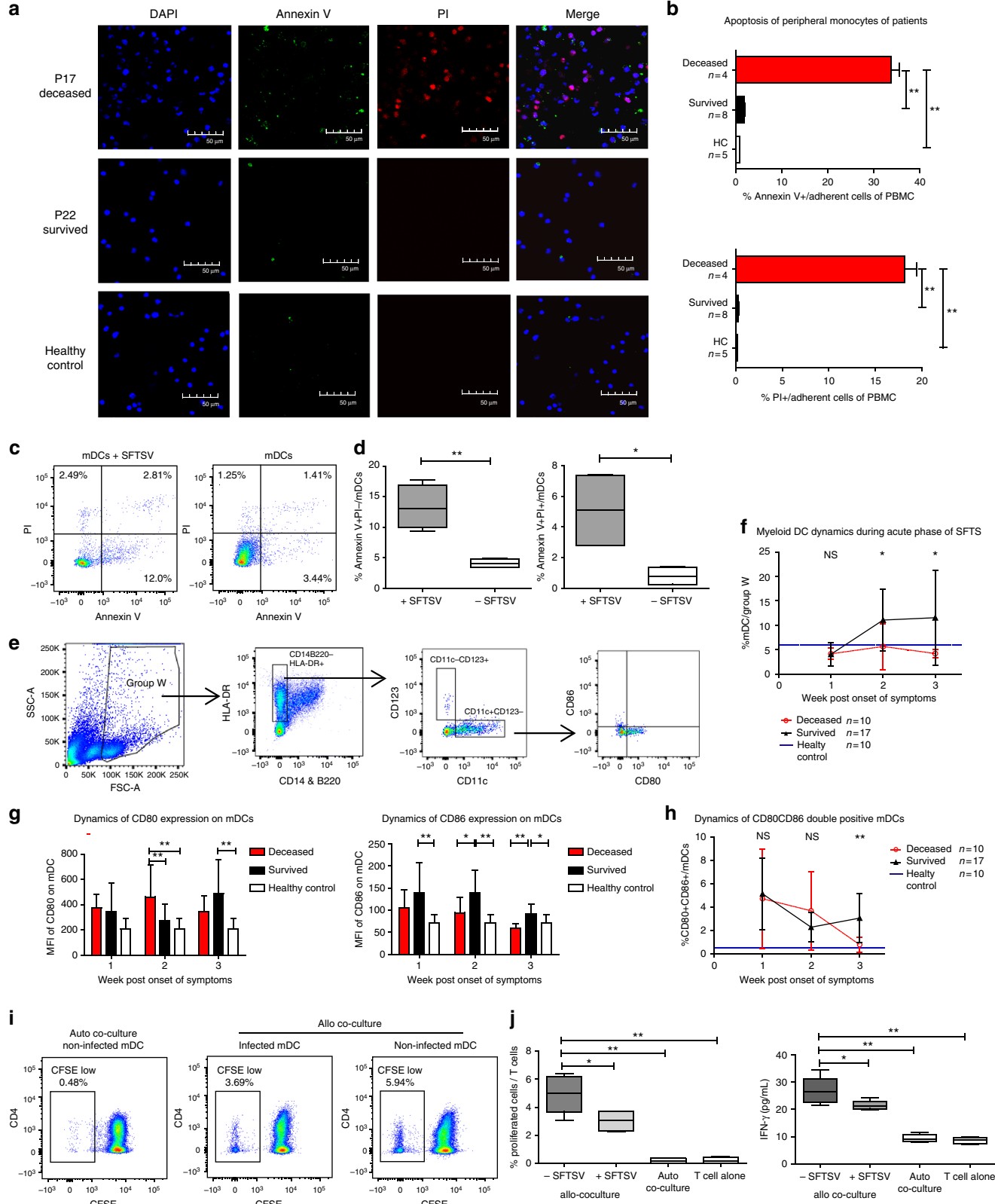

patients was mainly due to the expression of CD86, which may impact on their antigen presentation function. In order to further evaluate the antigen-presentation function of mDCs during SFTS virus infection, we isolated mDCs from peripheral blood of

healthy donors and performed allogenic DC/T-cell co-culture and stimulation assay using in vitro infection model[35–37] (Fig. 6i, j). When mDCs were infected with SFTS virus, the proliferation rate of T cells as measured by CFSE tagging and the level of IFN-γ in

**Fig. 6** Apoptosis of peripheral monocytes and differentiation of myeloid DC. **a** Apoptosis of peripheral monocytes in deceased (P17) and survived (P22) patients, and healthy control, as shown by annexin V and PI staining. The patient samples were collected in the 1st week post symptom onset. **b** The ratios of PI⁻ and annexin V-positive cells in total adherent cells of PBMCs were measured, and then compared among deceased, survived and healthy groups using one way ANOVA. **c**, **d** Apoptosis of SFTSV-infected mDC, as shown by annexin V and PI staining. Percentage of apoptotic cells (**c**) and statistic analysis of the annexin V⁺PI⁻ and annexin V⁺PI⁺ cells (**d**) in SFTSV-infected and uninfected mDCs ($n = 4$). Significant difference was calculated using one way ANOVA. **e** FACS gating strategy for defining mDCs and CD80/CD86 expression on peripheral monocytes. Group W was gated as a collective set of DCs and other lineage (+) cells as shown, including DCs, HLA-II⁺CD14⁺B220⁺ cells and a part of HLA-II⁻ cells. **f** Kinetics of peripheral mDC population during acute phase of SFTS. The means of mDC numbers of the 17 survived and 10 deceased patients was compared as the proportion of mDC in the upper population gated in FACS. Statistic analysis was conducted between the survived and deceased groups. **g**, **h** Comparison of MFI of CD80 or CD86 expression on mDCs and CD80⁺CD86⁺ mDC population among survived, deceased patients and healthy individuals for the 3-week period post symptom onset. Black triangles, red circles and blue horizontal lines represent means of survived, deceased and HC group respectively. **i** Representative results of T-cell proliferation in MLR assay, measured by FACS analysis of CFSE labelling. Proliferative cells were defined as CFSE-Low subsets in the labelled T cells. **j** Statistic analysis of T-cell proliferation in MLR assay. Triplicated assays were performed to the mDCs from the healthy donors (left). IFN-γ production in the co-culture was quantitated by ELISA (right). Significant difference was calculated using one way ANOVA. *$p < 0.05$, **$p < 0.005$, NS = no significance. Error bar represents the standard deviation

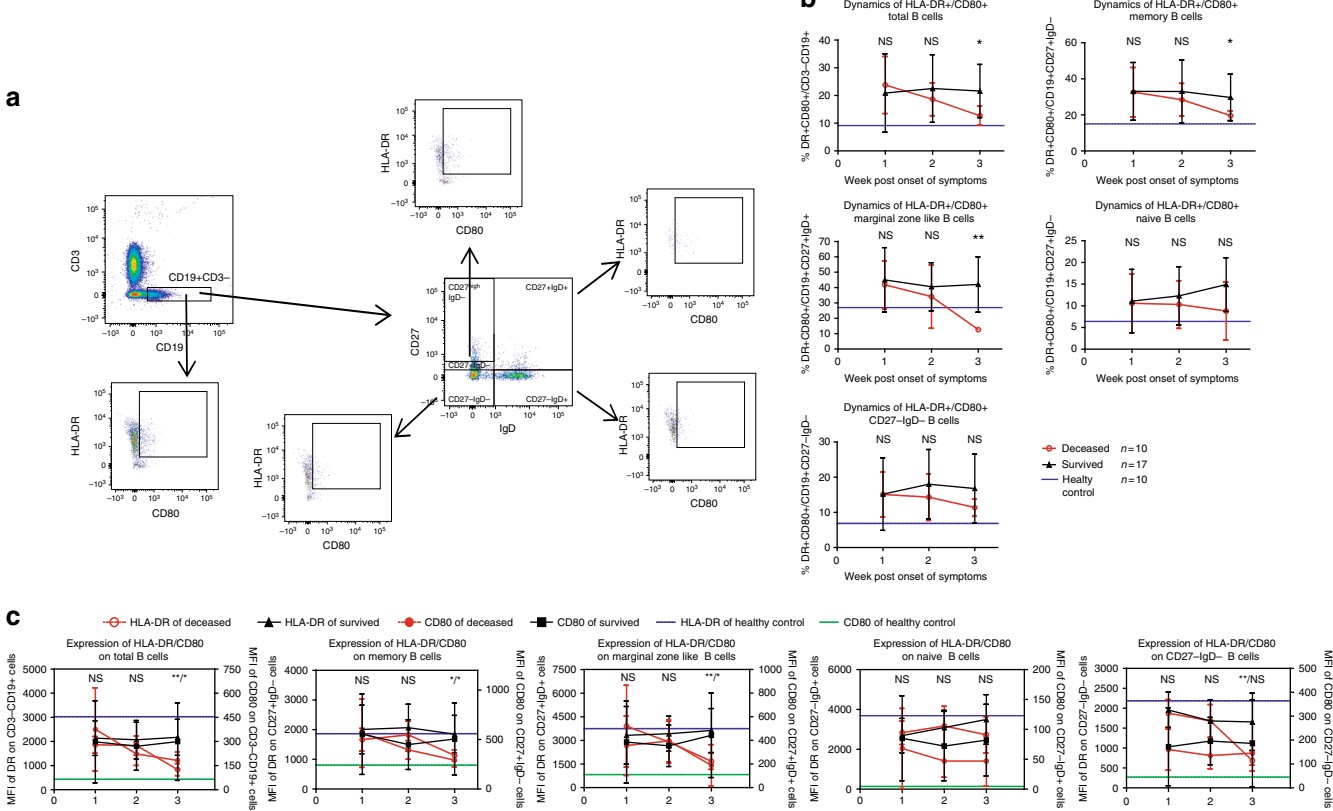

**Fig. 7** Phenotypic analysis of antigen-presenting B-cell subsets during acute phase of SFTS. **a** FACS gating strategy for the measurement of the expression of HLA-DR and CD80 on B-cell subsets. **b** Kinetics of the proportion of HLA-DR⁺CD80⁺ B cells in various B-cell subsets in the 3-week period post symptom onset among survived and deceased patients and healthy individuals. Black triangles, red circles and blue horizontal lines represent means of survived, deceased and HC group, respectively. **c** Kinetics of the expression of HLA-DR and CD80 on B-cell subsets as measured by MFI. All symbols in the graph represent means of the expression of HLA-DR or CD80 of different groups. Statistic analysis was conducted between the survived and deceased group.*$p < 0.05$, **$p < 0.005$, NS = no significance. Error bar represents the standard deviation

the supernatant were significantly lower than that of the mock infection, demonstrating that mDC activation of T cells could be inhibited by SFTS virus infection.

**Inhibited HLA-DR/CD80 on peripheral B cells in fatal cases**. B-cell antigen presentation, in cooperation with DCs, facilitated Tfh differentiation and GC formation[11]. Previous study revealed that a small subset of B cells mainly located at the T-cell B-cell border of lymphoid organs were involved in the process[38]. Therefore, we indirectly estimated antigen-presenting function of this B-cell

subset by calculating the expression of CD80 and HLA-DR on peripheral B cells (Fig. 7a). HLA-DR⁺CD80⁺ expression had been proved to be a key factor affecting the B-cell antigen presentation[39]. In the deceased patients, the HLA-DR expression on total B cells, MBs, MZ B and CD27⁻IgD⁻ B cells was significantly more down-regulated than that in the survived patients in the 3rd week (Fig. 7c), as shown by the dynamics of the HLA-DR⁺CD80⁺ B cells (Fig. 7b; Data of individual patients were shown in Supplementary Fig. 5). Meanwhile, the proportion of CD80⁺HLA-DR⁺ cells and the CD80 expression in total B cells, MBs

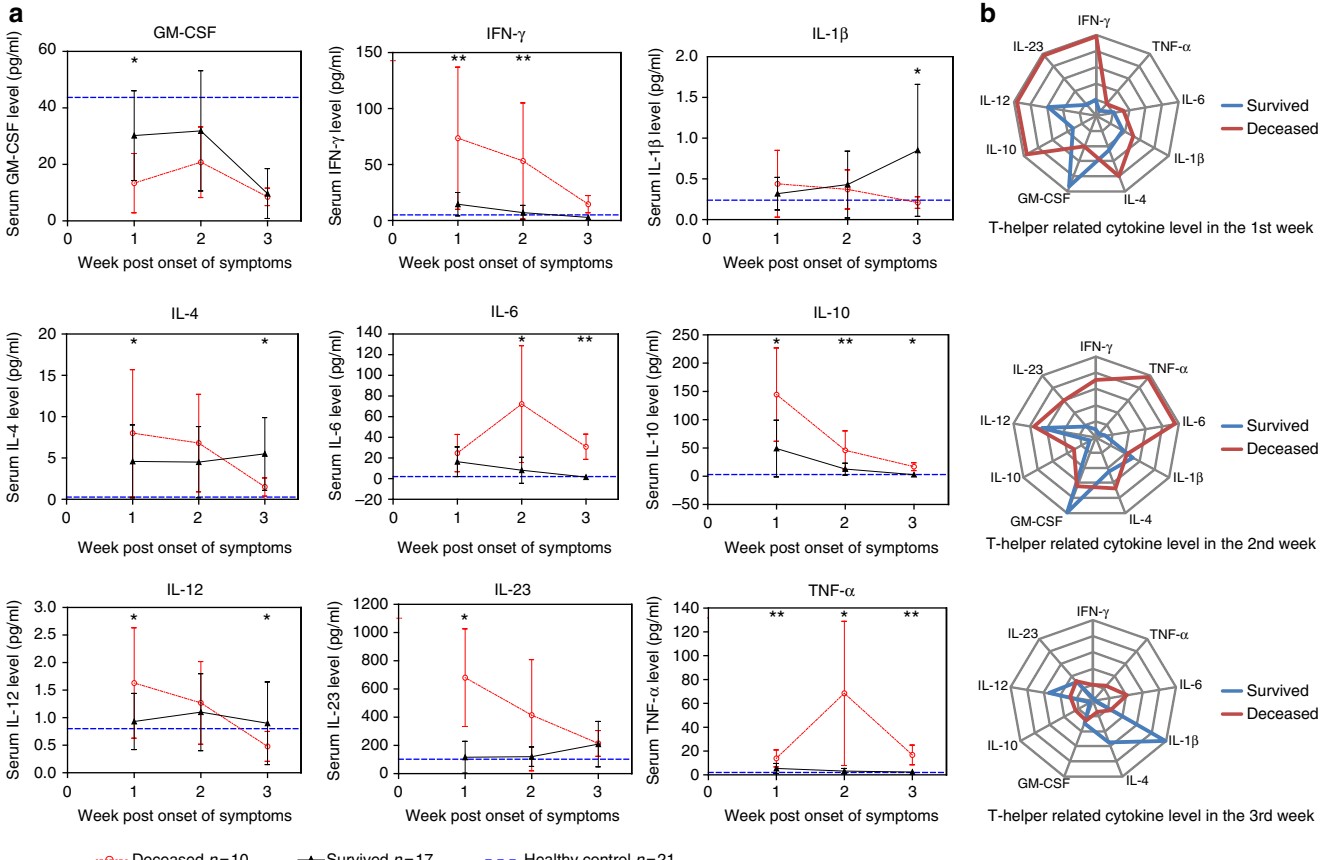

**Fig. 8** Dynamic modulation of serum T-helper-related cytokines in selected SFTS patients. **a** Kinetics of serum cytokines in the 3-week period post symptom onset and comparison of the level of each cytokine between survived and deceased patients. Black triangles, red circles and blue horizontal lines represent means of survived, deceased and HC group, respectively. Statistic analysis was conducted between the survived and deceased group. *$p < 0.05$, **$p < 0.005$, NS = no significance. **b** Radar plots comparing regulatory profiles of the cytokines between the survived and deceased groups are generated from **a** to indicate the dynamic changes of cytokines during the three-week clinical course. The radar periphery represents higher values of the cytokine expression. Error bar represents the standard deviation.

and MZ B cells but not in naive and CD27⁻IgD⁻ B cells differed between the deceased and the survived groups (Fig. 7b, c).

**Serum Th-related cytokine kinetics during acute infection.** Cytokines are important to the maturation and function of Tfh and DCs. We, therefore, determined relevant cytokines in SFTS patients' sera using a highly sensitive Milliplex chip (Fig. 8a). A radar plot analysis was presented in Fig. 8b to show the dynamic profile of the cytokine regulation during the three weeks of acute infection. In the 1st week, serum levels of IFN-γ, IL-23, IL-12 and IL-10 increased dramatically in the deceased patients, with IL-4 and TNF-α increased to a less extent as compared with these in the survived patients whose GM-CSF level was higher. In the 2nd week, IFN-γ, IL-23, IL-12 and IL-10 showed declining trend in the deceased group, but still had higher levels than those in the survived group except for IL-12. During the same period, IL-6 and TNF-α, two representative inflammatory cytokines, were markedly elevated in the deceased patients. Although nearly all cytokines measured in the deceased patients decreased further in the final week, as compared with those in the previous two weeks, IL-6 and TNF-α still maintained higher level than those in the survived patients. However, serum IL-1β and IL-4 in the survived patients markedly elevated accompanied by improved clinical course. Interestingly, the level of IL-21 in the sera of nearly all individuals was undetectable.

## Discussion

Since 2010, SFTS has been a persistent public health threat in the East Asia. Almost at the same time, another novel phlebovirus termed Heartland virus (HRTV) was reported in several states of America. HRTV infection shares many common clinical characteristics with those of SFTSV, including high fatality rates in hospitalized individuals[40]. Until now, the pathogenesis and the roles of humoral response in the infection are poorly understood.

Here we reported that the failure of virus-specific IgG response is a hallmark of the fatal outcome of SFTS, as shown by the absence of both serum IgM and IgG specific to NP, and the absence of IgG specific to Gn in the deceased patients. Similar phenomenon was reported in Ebola virus infection, though only in a few cases[41], but not in other virulent viral infections[42–47]. The absence of antibody response to SFTSV in fatal cases also illustrates that current serologic tests alone are insufficient for the diagnosis of the SFTSV infection, particularly in the severe cases. Neutralization assay confirmed the deficiency of functional humoral response in the fatal SFTSV infection, and implicated the potential roles of Gn-specific antibody in infection control. However, it is not clear if the absence of Gc reactivity reflects the poor immunogenicity of Gc or the Gc does not express native epitopes that can be recognized by serum antibodies as no reactive antibody is currently available.

Previous studies in Ebola and other virus infections reported that the magnitude of PB proliferation closely correlated with virus clearance[25,47]. However, phenotypic analysis of PBs by two

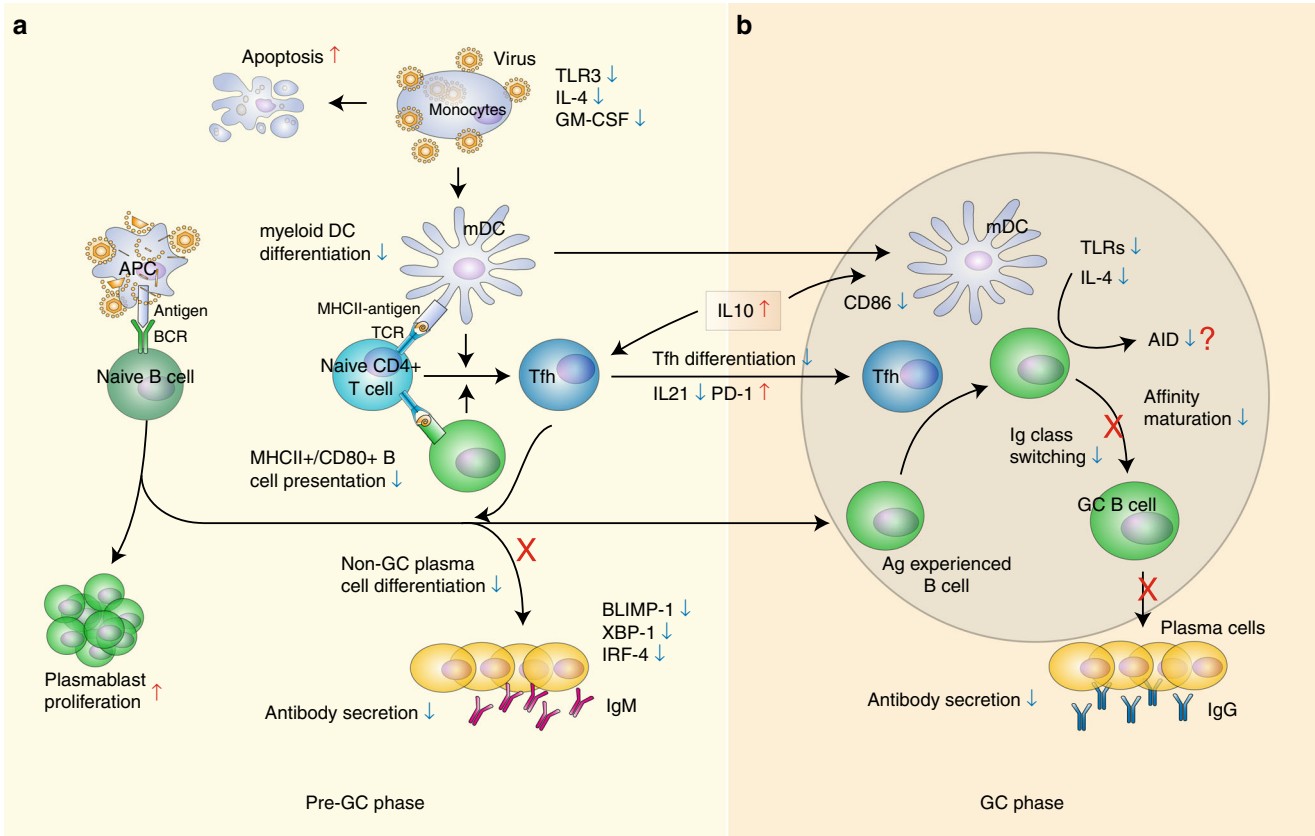

**Fig. 9** Schematic presentation of the mechanisms in the defective humoral response to SFTSV infection. **a** In the pre-germinal center (pre-GC) phase, persistent viremia primes large quantity of naive B cells into PBs and causes massive monocytic apoptosis, resulting in impaired mDC differentiation, accompanied by the downregulation of TLR3, IL-4 and GM-CSF. Due to insufficient MHC-II antigen presentation by mDCs and B cells, the differentiation and function of Tfh are inhibited, characterized by the decreased numbers and downregulation of IL-21 secretion. Additionally, the expression of PD-1 on Tfh, which act as an important inhibitory molecule of T-cell function, was observed to be persistently upregulated in deceased patients. The expression of three key transcriptional factors, BLIMP-1, IRF-4 and XBP-1, was down-modulated due to the lack of Tfh stimulation. PBs failed to differentiate into non-GC plasma cells and were unable to secret IgM antibodies. **b** In the GC phase, Ag-experienced B cells interact with cognate Tfh and mDC. However, the B-cell maturation is impaired due to the dysfunction of cognate Tfh and mDC, which, together with the downregulation of TLRs and IL-4, inhibited the induction of activation-induced cytidine deaminase (AID), resulting in the complete inhibition of immunoglobulin class-switch in the deceased patients. Red upward arrow and blue downward arrow denote up and downregulation, respectively

different gating strategies showed a consistent but unique phenomenon that PBs in the deceased patients proliferated to a larger extent than that in the survived patients, which is apparently in contrast to the weakened production of the virus-specific antibody. The negative intracellular staining of IgG and IgM in PBs and the absence of virus-specific IgG provides compelling evidence on the dysfunctional proliferation of PBs in the fatal SFTS cases. Furthermore, the inhibited expression of BLIMP-1, IRF-4 and XBP-1, critical for the maturation and antibody production of PBs, indicated that the CD27$^+$CD38$^+$ B subset in fatal SFTSV infection was only phenotypical but not functional PBs. This functional defect of PBs was likely attributable to the inhibition of monocytes and dysregulation of cytokines[48], which is under further investigation.

Class-switch recombination (CSR) is crucial for establishing effective anti-viral immunity, which is mediated by activation-induced cytidine deaminase (AID) in mature B cells[49–51]. Recently, the molecular mechanisms that control CSR have been elucidated[52,53]. A study by Pone et al. demonstrated that BCR-signaling synergizes with TLR-signaling to induce AID and immunoglobulin class-switching[52]. Feldman et al. found that the chromosome conformation of constant region (C$_H$) loci, which served as the target for CSR, was regulated in a cytokine dependent fashion in mature B lymphocytes[53]. Interestingly, in our

previous[8] and present studies, we observed that TLR3 signaling pathway and IL-4 secretion were both significantly inhibited in fatal SFTSV infection.

The dynamics of pTfh during acute viral infection in patients hasn't been previously reported, despite limited data on HIV and SIV infections in animal models[12,13,54]. We found that the pTfh peaked at the early phase of acute infection in survivors, then rapidly declined as the infection proceeded. This phenomenon can be explained that the antigen-experienced Tfh quickly migrates into the GCs to provide systemic T help for the establishment of the humoral response[55]. Similar phenomenon was observed in SIV-infected Rhesus Macaques, in which an early loss of splenic Tfh resulted in the deficiency of antigen-specific antibody production[12]. Therefore, we suggest that an early expansion of Tfh is critical for humoral response to SFTSV infection and for the control of viral pathogenesis. Although previous study demonstrated that ICOS played critical roles in protective humoral response against Influenza virus[30], ICOS on pTfh exhibited no significant difference between two SFTS patients' groups, indicating that it is not related to the failure of specific antibody response. However, the expression of PD-1, a crucial inhibitory molecule of T cells, was upregulated and maintained at a high level throughout the entire disease course in the deceased patients. Despite PD-1-mediated CD8$^+$ T-cell exhaustion has

been well addressed in chronic infections and cancer[56,57], its roles in acute viral infection, especially in the regulation of CD4[+] T subsets are poorly understood. A recent study reported that high percentages of CD4 and CD8 T cells expressing CTLA-4 and PD-1 were the unique immune signature in fatal Ebola virus infection[29]. Our observation illustrated that the lethal SFTSV infection might share this "unique" immune characteristic with Ebola virus infection. Notably, anti-PD-1 immunotherapy has been shown to be effective and safe in controlling persistent HCV infection in chimpanzees[56]. Therefore, PD-1 may serve as a potential therapeutic target in lethal SFTSV infection.

In our previous study, we found that mDCs were impaired in the fatal cases[8], consistent with a recent report by Zhang et al.[58]. Since monocytes were reported as the target cells of SFTSV[18,59], we investigated the apoptotic status of peripheral monocytes of SFTS patients and found that monocytes underwent severe apoptosis and necrosis even at the early stage of the fatal infection. This observation is consistent with our previous work[8], in which we found that the gene expression of IL-6 and CD40L in the monocytes was markedly inhibited with the development of disease severity. CD40L signaling and IL-6 have been reported to confer the apoptotic resistance to a variety of stress inducers, including virus infection[60]. Moreover, significantly elevated apoptosis rate of mDCs shown by in vitro infection assay confirmed the direct cytopathic effect of SFTSV on mDCs.

Both phenotypic and functional analysis revealed that SFTSV infection could impair the antigen-presenting function of mDCs. In fact, down-modulation of the co-stimulatory molecules on DCs was also observed in other viral hemorrhagic fever[61,62]. An in vitro study demonstrated that in HIV infection the frequency of CD86[+] mDC approached 100% in all conditions, in contrast to CD80 expression on mDCs that was almost completely inhibited[63]. However, the regulation of these two co-stimulatory molecules in SFTS exhibited an entirely distinct profile. Meanwhile, phenotypic analysis of peripheral B cells demonstrated inhibited expression of both HLA-DR and CD80 on memory and MZ B cells, indirectly indicating the impairment of antigen-presentation function of B cells. After all, because the transcriptional and phenotypic similarities between peripheral and follicular B cells involved in antigen-presenting process are still poorly understood, the profiling of co-stimulatory molecules in the peripheral B cells would only serve as an indirect evidence for this process. Therefore, animal model study should be performed to further elucidate the role of B-cell antigen presentation in the pathogenesis of SFTSV infection in further investigation.

Two unique characteristics of cytokine regulation were observed in the fatal SFTS cases. One is that IL-10, IL-12 and IL-23, all immune-suppressive cytokines, were drastically elevated at the early phase of infection[64,65], followed by a subsequent inflammatory cytokine storm leading to the cytokine-mediated organ damages. IL-10 has been shown to suppress GC formation by affecting Tfh cell function, in part by altering the expression of transcription factors BCL-6 and BLIMP-1[66]. Additionally, earlier studies showed that IL-10 down-regulated MHC-II expression and co-stimulatory molecule B7, which is consistent with our observations in the present study[67,68]. The expression of IL-4 and GM-CSF, both absolutely required for the differentiation and maturation of DCs[15,16,69], was markedly down-modulated in both monocytes and lymphocytes, especially in severe and fatal cases[8]. These observations strongly suggested that these cytokines are involved in the pathogenesis of SFTSV infection.

In summary, we demonstrated that a complete absence of virus-specific IgG response to NP and Gn was associated with the fatality of SFTS and the impaired humoral response was the result of a global disruption of B-cell immunity. We propose that during

fatal SFTSV infection (Fig. 9), large quantity of naive B cells are continuously primed into PB cells by the presence of persistent viremia. However, due to defective mDC maturation resulting from massive monocytic apoptosis and insufficient IL-4 and GM-CSF, the differentiation of naive T cells into Tfhs is impeded. Lack of cognate Tfh and MHCII antigen presentation by mDCs and B cells, the initially primed antigen-experienced PB cells fail to mature and undergo immunoglobulin CSR, which accounts for a complete absence of virus-specific IgG response to NP and Gn, in the deceased patients. The current study also points to the roles of NP- and/or Gn-specific antibody response in the control of viral pathogenesis and shed light to the potential target for vaccine or therapeutic antibody development.

## Methods

**Patients and clinical samples.** Clinical specimens were obtained from 30 SFTS patients admitted into Nanjing Drum Tower Hospital from April to September in 2016. The cohort consists of 11 deceased and 19 survived patients who were confirmed of SFTSV infection by RT-PCR. Serum specimens were stored at −80 °C until use. Fresh PBLs were utilized when needed and, otherwise, cryogenically preserved until use. Informed consent was obtained from all subjects, in accordance with the Declaration of Helsinki, and the research was approved by the Ethics Committee of Nanjing Drum Tower Hospital.

**ELISA analysis of serum antibodies specific for NP and Gn.** Serum titers of IgM and IgG to SFTSV nucleocapsid protein (NP) were measured using commercial ELISA kits (Sinosbio, China) following the manufacturer's instructions. Serum specimens were diluted to serial concentrations of 1:20, 1:40, 1:80, 1:160 and 1:320, and analyzed in duplicate. Human serum IgG specific for SFTSV Gn was quantitated by ELISA. 293T expressed and purified Gn was diluted in antigen-coating buffer to a predetermined concentration in 96-well plates. The plate was incubated at 4 °C overnight. Serum was added to the plate at 1:100 and allowed to bind Gn in triplicate at 37 °C for 1 h. Anti-human IgG conjugated to HRP (Abcam, USA) was used as the secondary antibody. The optical density of the reaction was measured at $OD_{405}$.

**Quantitative RT-PCR.** The viral load in patients' serum was determined on an ABI 7500 Real-time PCR system (Life Tech, USA). The primer set was as below: forward primer: 5′-GGGTCCCTGAAGGAGTTGTAAA-3′, reverse primer: 5′-TGC CTTCACCAAG ACTATCAATGT-3′, probe: 5′-FAM-TTCTGTCTTGCTGGC TCCGCGC -BHQ1–3′. The relative gene expression of BLIMP-1, IRF-4 and XBP-1 in sorted CD27[+]CD38[+] B cells, and IL-21 in sorted pTfh were determined on QuantStudio™ 6 Flex Real-Time PCR Systems (Life Tech, USA) using TaqMan® Gene Expression Assays (Thermo Fisher, USA) according to the manufacture's procedure. The quantitative level of mRNA was normalized to β-actin using the cycle threshold (Ct) method ($2^{-\triangle\triangle Ct}$ method), then compared in $\log_2$ value.

**Expression of soluble SFTSV Gn and Gc.** Amino acid sequence of Gn was analyzed by TMHMM Server v.2.0. Both the hydrophobic N- and C-termini of Gn or Gc were removed for a soluble expression. The truncated sequence of Gn expands from amino acid residue $D_{20}$ to $C_{447}$, whereas the truncated sequence of Gc expands from amino acid residue $C_{563}$ to $N_{1026}$, based on Genbank accession number: YP_006504094.1 for the membrane glycoprotein polyprotein [SFTS virus HB29]. The expression fragment was constructed as Kozak sequence—signal peptide: MGWSCIILFLVATATGVHS—Gn $D_{20}$–$C_{447}$—His tag—stop codon for Gn and MGWSCIILFLVATATGVHS—Gc $C_{563}$~$N_{1026}$—His tag—stop codon for Gc. The DNA sequences were synthesized by Y-Clone Ltd. (Suzhou, China), and were cloned into pVAX1 plasmid to generate pVAX1-GnRFC or pVAX1-GcRFC expression vector. We transfected 293F or CHO cells and purified Gn and Gc, respectively, by Ni[+]NTA beads (Smart-Lifesciences, Suzhou, China) following the manufacture's protocol. Gn and Gc were identified by western blot analysis, and anti-His antibody (Santa Cruz, USA) served as the primary antibody at a dilution of 1:300.

**Western blot analysis.** Serum IgG and IgM to Gn and Gc were analyzed by western blot. Briefly, Gn or Gc was separated by SDS-PAGE and transferred to PVDF. The PVDF was stained overnight at 4 °C with serum diluted at 1:300 or 1:100 for IgG or IgM detection, respectively. The membranes were then incubated with anti-human IgG or IgM secondary antibody conjugated to HRP (Abcam, USA) at a dilution of 1:10000 or 1:5000, respectively, for 4 h. The immunoreactive bands of IgG and IgM were detected by Hyperfilm ECL (Pierce, USA) and FluorChem FC2 system (Alpha Innotech Corporation St. Leonardo, CA, USA), respectively.

**Viral stocks**. SFTSV was isolated from an acutely infected patient's serum and propagated in Vero cells.. The supernatant was diluted into tenfold serial dilutions with DMEM, which was used to inoculate Vero cells in 12-well plates. The cells were transferred to glass cover slides at 8 or 18 h post infection, air dried, fixed, and permeabilized with 4% paraformaldehyde and 0.1% Triton X-100. Vero cells were stained with camel Gn-IgG positive serum at 1:500, followed by Alexa Fluor 488-conjugated rabbit anti-camelid VHH antibody (Genscript, China) at 1:1000. Infectious virus titers ($TCID_{50}$/ml) were calculated according to the Reed and Muench method. $1 \times TCID_{50}$/ml aliquots were stored at $-80\,^{\circ}C$. The viral load of $1 \times TCID_{50}$/ml was quantitated at about $10^5$ copies/ml by RT-PCR.

**Neutralization assays**. Camels were primed with Gn prepared in CFA, boosted by Gn in IFA every two weeks, and the sera were collected at the end of the 8th week. Serum antibodies were identified by ELISA and WB analysis. Serum specimens were inactivated in $56\,^{\circ}C$ for 30 min. The neutralization activity of the sera from a convalescent SFTS patient (30 d post symptom onset), a fatal patient (one day before death) and a Gn-immunized camel was determined by the focal reduction neutralization test (FRNT). The number of the infected cells was visualized and calculated using Olympus Fluoview FV3000 (Tokyo, Japan). The reduction of the infected foci was then compared with that in the control (no serum). Serum specimens in serial dilutions of 1:20, 1:40, 1:80, 1:160, 1:320, 1:640 and 1:1280, were mixed with equivalent volume of $1 \times TCID_{50}$/ml viral aliquots separately. The mixtures were incubated for 1 h at $37\,^{\circ}C$, transferred to Vero cells in glass slides covered 12-well plates and incubated for 2 d. The cells were stained with camel Gn-IgG positive serum at 1:500, followed by Alexa Fluor 488-conjugated rabbit anti-camelid VHH antibody (Genscript, China) at 1:1000. 50% FRNT values for each serum dilution were determined by Graphpad Prism 5 software (USA).

**Apoptosis analysis of SFTS patients' PBMCs**. PBMCs were isolated by density gradient centrifugation using Ficoll-Paque Plus (GE Health Bioscience, Sweden) according to the manufacture's protocol. $4 \times 10^5$ PBMCs were cultured in RPMI-1640 supplemented with 10% heat-inactivated FBS in 24-well plates at $37\,^{\circ}C$, 5% $CO_2$ for 3 h. Cultural medium was gently removed and replaced with 5ul of annexin-V-FITC and PI working solution (Beyotime Biotechnology, China). The plate was incubated at room temperature in the dark for 10 min and washed 3 times with Binding Buffer. Images were acquired using Olympus Fluoview FV3000 (Tokyo, Japan).

**Tfh stimulation**. About $10^6$ PBMCs from SFTS patients were cultured with RPMI-1640 supplemented with 10% heat-inactivated FBS and stimulated with 2ul Leukocyte Activation Cocktail (BD bioscience, USA) for 3 h at $37\,^{\circ}C$. The cells were then collected for surface and intracellular staining for flow cytometric analysis as described below.

**Flow cytometric analysis**. All antibodies and reagents used in flow cytometry analysis were purchased from BD Biosciences, USA, and the antibodies were grouped into six panels for the phenotypic analysis of B cells (A, B), PB and MB cells (C, D), mDC (E) and Tfh (F), respectively. Red blood cells were removed by lysing in Red Blood Cell Lysis Buffer. Antibody combination in each panel was as follow: Panel A (CD3-PerCP-Cy5.5, CD19-FITC, CD27-APC, IgD-PE, HLA-DR-PE-Cy7, CD80-APC-H7), Panel B (CD19-FITC, CD27-APC, IgD-PE, IgM-PE-Cy7, IgG-APC-Cy7, CD21-PerCP- Cy5.5), Panel C (CD3-PerCP-Cy5.5, CD19-FITC, CD27-APC, CD38-PE-Cy7, IgG-APC-Cy7, IgM-PE), Panel D (CD3-PerCP-Cy5.5, CD19-FITC, CD27-APC, CD38-PE-Cy7, IgA-APC-Vio770, IgD-PE), Panel E (CD14/B220-APC-Cy7, CD11c-PE, CD123-APC, HLA-DR-PE-Cy7, CD80-FITC, CD86-PerCP-Cy5.5) and Panel F (CD3-PerCP-Cy5.5, CD4-APC-Cy7, ICOS-AlexaFluor647, CXCR5-BB515, PD-1-PE-Cy7, IL-21-PE). Intracellular staining of IgG, IgM and IL-21 was performed at room temperature for 30 min following cell fixation and permeabilization with FIX/PERM buffer. Immune phenotyping was carried out on a 6-laser Aria II (BD Biosciences, USA) Multiparametric flow cytometer. Data were analyzed using FlowJo version 9.2 (TreeStar, USA).

**B-cell Elispot**. Peripheral blood mononuclear cells (PBMCs) were prepared from fresh heparinized peripheral blood specimens by density gradient centrifugation using a Ficoll cushion. After washing, PBMCs were re-suspended at $2 \times 10^6$ PBMC/ml in RPMI-1640 medium (Gibco, Invitrogen, Paisley, UK) supplemented with streptomycin, glutamine, and 10% fetal bovine serum (FBS). To stimulate B cells, 1 μg/ml R848 and 10 ng/ml recombinant human (rh) IL-2 were added (Mabtech, Nacka Strand, Sweden). For in vitro Gn stimulation, PBMCs were incubated with 2 μg/ml Gn, 1 μg/ml R848 and 10 ng/ml IL-2. The cells were subsequently incubated for three days at $37\,^{\circ}C$, 5% $CO_2$. Stimulated cells were washed in RPMI-1640 prior to plating. Sterile 96-well Multiscreen-IP filter plates with a PVDF membrane (Millipore Corp., Bedford, MA, USA) were coated with anti-human IgG (15 μg/ml, Mabtech, Nacka Strand, Sweden) or Gn (10 μg/ml) overnight at $4\,^{\circ}C$. Numbers of total $IgG^+$ secreting cells or Gn-specific B cells were determined by wells coated with anti-human IgG or Gn, respectively. Wells with no antibody coating were used as negative controls. Plates were washed with sterile PBS and blocked with RPMI-1640 medium/10% FBS for 2 h at room temperature. 5000 or 200,000 pre-activated cells were added to IgG- or Gn-coated wells in duplicates, respectively, and incubated at $37\,^{\circ}C$, 5% $CO_2$ for 18 h. The frequency of antibody-secreting cell was measured after incubating the cells with biotin-labeled anti-IgG mAb (Mabtech, Nacka Strand, Sweden) followed with horseradish peroxidase (HRP)-conjugated streptavidin (Mabtech, Nacka Strand, Sweden). Image analysis was carried out on an automated ELISpot image analyzer (Cellular Technology Limited, Hong Kong, China).

**CD4$^+$ B-cell helper assay**. pTfh cells ($CD3^+CD4^+CXCR5^+ICOS^+PD1^+$) and naive B cells ($CD3^-CD19^+CD27^-$ $IgD^+$) were sorted from the PBLs from patients or healthy donors (HC) by FACS. The pTfh cells were stimulated with cell stimulation cocktail for 6 h, completely washed, and labelled with CFSE. Naive B cells were used as a control. The stimulated pTfh were subsequently co-cultured with autologous naive B cells at a 1:1 ratio for 3 days. Flow cytometry was used to determine the proliferation, IgM and CD38 expression of the co-cultured naive B cells.

**mDC apoptosis assay and allo-MLR assay**. Fresh PBMCs were isolated from healthy donors. mDCs were isolated using CD1c (BDCA-1)+ Dendritic Cell Isolation Kit (Miltenyi Biotec, Germany). The mDCs were cultured in the complete medium containing GM-CSF and IL4 (20 ng/ml, R&D systems) and stimulated by LPS (0.1 μg/ml, Sigma-Aldrich) before use. To infect, a portion of the mDCs were pulsed with SFTS virus (moi = 1) for two hours and cultured for 7 days. For apoptotic analysis, cultured mDCs were measured in duplicate by flow cytometry using FITC-annexin V apoptosis detection kit with PI (Biolegend, China). For allo-MLR assay, the SFTSV-infected or uninfected DCs were harvested and then mixed with CFSE-labelled allogeneic $CD3^+$ T cells at a 1:10 ratio of DCs to T cells for allogeneic mixed lymphocyte reaction (Allo-MLR). The DC/T-cell coculture was carried out in complete medium containing IL-2 (20U/ml, R&D systems) for 3 days. The proliferation of CFSE-labelled $CD3^+$ T cells in allo-MLR was analyzed by flow cytometry.

**Multiplex ELISA**. Serum concentrations of selected cytokines were measured using Milliplex® Map Human High Sensitivity T-helper Cells Magnetic Bead Panel (Merck Millipore, USA) on a Luminex platform MAGPIX$^{TM}$ (Luminex, Austin, USA). The analysis was performed in accordance to the manufacturer's instructions.

**Statistics**. Continuous variables were reported as means ± SD. Data were analyzed by SPSS 11.0 software. Group means were compared using ANOVA or independent-samples $T$ test. Pearson's correlation tests were used to measure the strength of association between variables.

**Data availability**. All the data and materials used for this article are available from the authors on request. Datasets of this paper have been deposited in Open Science Framework under accession osf.io/9bjmd.

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

## Acknowledgements

We thank Mr. Yuxuan Fu for helping the graphics. This study was supported by The National Key Research and Development Program of China from the Ministry of Sciences and Technologies (2016YFC1201000), The Major Research and Development Project (2018ZX10301406), Nanjing University-Ninxia University Collaborative Project (2017BN04) and Fundamental Research Funds for the Central Universities (021414380341, 021414380432).

## Author contributions

P.S. participated in experimental design, performed experiments, conducted data analysis and wrote the manuscript; N.Z. participated in experimental design, performed experiments and data analysis; Y.L. performed FACS analysis, Luminex analysis of cytokines and PCR; C.T. performed B-cell Elispot analysis; X.W. and X.M. conducted construction of Gn/Gc expression system, expressed and purified the proteins and performed western blot analysis; D.C. performed apoptosis analysis and western blot; X.Z. performed western blot analysis; G.W. performed Luminex analysis of cytokines and PCR; H.W. carried out clinical sample processing and FACS analysis; Y.Z. and S.L. collected clinical samples and relevant clinical data; C.W. performed data analysis; Z.W. supervised the research project, carried out experimental design, wrote and edited the manuscript and provided funding and resources except clinical samples. P.S. and N.Z. contributed equally.

## Additional information

**Competing interests:** The authors declare no competing interests.

