## [Peer Review File · Nature Communications]

Reviewers' comments:

Reviewer #1 (Remarks to the Author):

Severe fever with thrombocytopenia syndrome (SFTS) is a severe infectious disease caused by the SFTSV virus. In some areas of the world, the disease is associated with a 10-30% frequency of fatality. The reasons why most patients survive, but 10-30% succumb to the disease is poorly understood. Previous studies mainly focussed on innate immune cells. In the present manuscript, it is shown that there is a striking lack of an anti-virus IgM and IgG response in all deceased patients (except a transient anti-Gn IgM antibody response in some deceased patients), whereas all patients who survived showed strong IgM and IgG responses against viral proteins. This is associated with a number of other immunological differences between patients who resolved the infection or died, namely i) persistent high virus titres in deceased patients compared to virus clearance in surviving patients, ii) strongly reduced frequencies of naive B cells, iii) increased frequencies of plasmablasts in the deceased patient group, iv) increased apoptosis of monocytes in the deceased patient group, v) decreased expression of MHC class II and costimulatory receptors on B cells from deceased patients, vi) alterations in TFH cells in the peripheral blood between the patient groups, and vii) alterations in cytokine levels between the patient groups.

Main criticism

1) In figure 3, only the relative frequencies of the various B cell subsets are presented, but it remains unclear, which subsets show changes in total numbers between the groups. For example, is there really a dramatic loss of naive B cells in the deceased patients, or is the lower frequency only due to higher number of other B cells, with total number of naive B cells actually unchanged? Thus, the total numbers of B cells should also be presented.

2) At first glance it is unexpected and may appear even contradictory that the deceased patients have no specific antibody response, but higher frequencies of plasmablasts. This may perhaps be explained by a general B cell activation towards plasmablast differentiation in the absence of a functional specific response. However, as this is a central issue, the plasmablasts need to be better characterized. It is strange that in the deceased patients about 70% and in the surviving patients 30% of plasmablasts are IgM-IgG-. A plasmablast without strong Ig expression is highly questionable. Perhaps, there is an induction of specific class-switching to IgA? There are also so-called IgD-only B cells and plasmablasts/plasma cells in the human that produce IgD in the complete absence of IgM. Can they play a role here? Moreover, the definition of the cells as plasmablasts relies only on the staining pattern CD27-high, CD38-high. Perhaps, in the acute viral infection some activated B cells transiently upregulate CD27 and CD38 without being plasmablasts? Thus, it is recommended that the CD27⁺⁺CD38⁺⁺ cells are also stained for IgA and IgD, and that the cells are analyzed for key plasmablast/plasma cell markers (upregulation of transcription factors BLIMP1, IRF4, XBP1) by FACS or RT-PCR.

3) One of the peripheral blood B cell subsets that showed a significant difference between the two patient groups are CD19⁺CD27⁻IgD⁻ B cells. In healthy individuals, it is well known that these are primarily IgG⁺CD27⁻ B cells, which account for 10-20% of all IgG memory B cells. However, because of the massive B cell compartment alterations present in the patients with acute virus infection, it remains uncertain what the cells defined by lack of CD27 and IgD really are. Are they indeed IgG⁺ memory B cells? Can it be excluded that they are, for example, immature B cells (IgM⁺IgD⁻CD27⁻), released from the bone marrow, or that they are polyclonally activated naive B cells that downregulated IgD upon stimulation? It would likely be sufficient to perform flow cytometric analysis in a few patients and healthy controls to stain the IgD⁻CD27⁻ B cells also for IgG and IgM to clarify their identity better. CD21 could be included to distinguish immature from unusual mature B cells, in case many of them are IgM⁺.

Minor points

a) The complete lack of an anti-humoral immune response against the virus in those patient who died is impressive. Some of the other alterations described may explain or contribute to the failed immune response. However, the authors need to consider that these are so far only correlations. Some of the interpretations need to be tuned down.

b) In figure 3B, the horizontal lines above the columns should not extend over the healthy controls, because they are distinct individuals that do not belong the patient groups.

c) On page 10, line 269, it is mentioned that an anti-B220 antibody was used to eliminate B cells from a DC isolation procedure. However, whereas B220 is indeed used as a pan-B cell marker in the mouse, this marker (recognising CD45R) is expressed only by a subset of human B cells. Please explain.

d) In figure 5, necrotic cells are defined as PI+ cells, and apoptotic cells as annexin V+ cells. However, one typically sees annexin V+PI- and annexin V+PI+ cells. How were the double positive cells grouped? As necrotic or as apoptotic? This needs to be better described.

e) At least in the pdf file available for review, the immunofluorescence pictures in figure 5A are very dark and hardly interpretable. The figure should be improved.

f) The Discussion is with nearly six pages very long. It would profit from shortening and focussing.

Reviewer #2 (Remarks to the Author):

The manuscript by Song et al showed the association of fatal SFTSV cases with a suppression of virus specific antibodies due to a lack of class switching in the plasmablast cells (PB). The authors attributed this phenomenon as the causative reason for disease aggravation and mortality, and attempted to define a mechanism that suggest a disruption of affinity maturation involving T_H, B cells and myeloid DC (mDC) activity in the lymph node or spleen. While the phenomenon of virus specific antibodies suppression is convincing, the experiments conducted to link this to dysfunctional T_H, mDC and B cells were loosely connected. Furthermore, the complex pathway that occurs within the secondary lymphoid organs and not the blood proposed in this study would require more than the surveillance of immune markers on the cells. These data should be supplemented by more conclusive functional in vitro assays or in vivo models.

Comments

1. The introduction contains too much irrelevant information and could be shortened.

2. Wrong reference cited for line 104 – 106. That study used non-monocyte derived cell lines for infection.

3. The following points should be addressed for Fig 1 and Fig 2:

a. To confirm that antibodies in the survive patients play a role in the clearance of virus and protection, functional neutralization assays with the patients' serum samples should be performed. Otherwise this remains an association and its impact should not be over claimed.

b. Fig 1, the profiling of NP specific antibody (Ab) is by ELISA and profiling of Gn specific Ab is by western blot. Data of Gn by ELISA is left as unpublished data. To provide better consistency, it would be better to present the ELISA data of Gn together with ELISA data of NP in the main figure and shift the data of Gn western blot to the supplementary with the data from Supp Fig 1.

c. Authors should perform clustering of the data in color chart of the ELISA data. As it stands, the current figure makes it very difficult to read the data.

d. Fig 2B, the way the graphs are plotted is not meaningful to observe correlation. Correlation curve plotted in the manner of Fig 2C is sufficient.

e. Fig 2C, the number of data points shown do not correspond to the number of data points collected. It is not clear how the authors selected the data to generate the correlation curve. It would be more meaningful to convert all the data points collected in Fig 2B into the correlation plot with just the viral load versus the Ab titre.

4. The following points should be addressed for data presented in Fig 3 and Fig 4:

- a. It is possible that the disease could induce leukopenia in patients and diminish the overall B cells numbers in the blood. Authors should provide data of blood counts that includes leucocytes and granulocytes. In addition, the numbers of total B cells should be compared between groups. Without this information, data in Fig 3C is not meaningful, as it is possible that while the % of a specific subset relative to total B cells is the same between survived and deceased groups, the actual numbers could be vastly different due to overall reduction of B cells.
- b. The rationale of having 2 different gating strategies for PB and MB was not clearly explained. Authors should consider including the CD38 marker into the analysis presented in Fig 3. In this way, the analysis of B cells subset definition could be standardized.
- c. Fig 4, the ELISPOT was done using total B cells. Authors should consider doing the ELISPOT with sorted PB to demonstrate the lack of virus specific Ab secreting PB cells.
- d. In lines 235 – 239, authors concluded that the observation of increase proportion of IgM-IgG- in the deceased group meant this subset is responsible for abrogated serological response. This is similar to saying that this subset of cells is immunosuppressive, which is not true. In addition, the comparison of this subset was done in % relative to total B cells. Data of % relative to total leucocytes would be more meaningful.

5. The following points should be addressed for data presented in Fig 5:

- a. Author presented 2 effects to the mDCs: apoptosis/necrosis and the suppression of co-stimulatory molecules that could suppress Ag presentation. The manuscript would flow better if the text from lines 258 – 283 is separated into 2 sub-sections based on these 2 effects.
- b. Regarding the apoptosis assay, authors allowed the PBMCs to adhere and subsequently performed staining on all adherent cells with the assumption they are monocytes. This is incorrect. The authors would need to either proof that this method produces homogenous monocytes in the adherent cells or add in markers to confirm the identity of the cells undergoing apoptosis. A more direct alternative method would be to track apoptosis using an annexin V kit by flow cytometry using the same gating strategy used for the mDCs.
- c. In the second part, authors profiles the co-stimulatory molecules and proposed that the Ag presentation of mDCs is impaired. This is inadequate and more direct function assays would be required to drive this point. Particularly, authors should sort out mDCs from the 2 groups of patients, and assess their ability to present Ag to T cells either by CFSE tagging to the T cells or by ELISOT assay.
- d. Lastly, an important fundamental concept must be addressed. Myeloid DC (also known as conventional DC) is derived from CDP and not monocytes.

6. Fig 6, authors proposed that B cells Ag presentation to T_H is vital for T_H expansion and the alteration of CD80 and HLA-DR disrupted this process. This unique process happens exclusively in the lymph nodes and spleen in a small percentage of B cells within the interfollicular zone². It is questionable if the profiling of co-stimulatory molecules in the peripheral B cells would be sufficient to drive this conclusion. Direct immunofluorescent staining of B cells and T_H co-localization in the interfollicular zone would be required to show the abrogation of this interaction.

7. Fig 7, where authors argued the loss of function T_H in deceased patients would need to be further addressed:

- a. Representative pseudoplot provided highlighted only 2 IL-21 positive staining events. This is too low to be conclusive if we consider the noise to actual signal ratio of a generic flow cytometry machine. In view of this, authors should consider using IL-21 gene expression from sorted pT_H cells.
- b. Following the reference cited by the authors, functional role of pT_H in patients was shown by

their ability to induce phenotypic changes in naïve B cells through co-culture system. They should consider using a similar system to assign a function difference to the pTfH beyond the profiling of IL21.

8. Lastly, it would make better sense to change the order of some of the data. Particularly, pTfH data should come after showing Ab suppression, followed by the B cell Ag presentation and mDC data which explain the change in TfH.

Reviewer #3 (Remarks to the Author):

In this manuscript, Song and colleagues identify the defective serological response that correlated with fatal SFTSV infection. By focusing their study on surviving and deceased patients following SFTSV infection, the authors show that deceased patients lacked a crucial IgG mediated response to viral proteins (nucleoprotein and glycoprotein) due to a B cell class switch failure. Additionally, they characterized severe apoptosis of peripheral monocytes in patients with the fatal outcome when compared to surviving patients or healthy controls. Finally, they performed cytokine profiling of all patients and found a correlation in severity with increased levels of serum IFN- γ , IL-10, IL-6, and TNF- α . Overall, this data provides an in-depth analysis of the B-cell mediated deficiencies that lead to SFTS severity, which has not been shown before in the field.

This manuscript could be improved by making revisions in response to the following minor comments.

Minor Comments:

The rationale for this manuscript is well founded based on a previous study by the same authors in 2017, where they found the reduction of both IFN- β and IL-1 β in patients that have a severe or fatal outcome of the disease. Furthermore, they observed a reduction of myeloid dendritic cells and an overall suppression of TLR3 in the primary targets for SFTSV, myeloid dendritic cells, which are primary targets for SFTSV infection, and an overall suppression of TLR3 in DCs. Hence, the authors postulated that antigen presentation was inhibited in patients that had a fatal outcome following SFTSV infection. While the premise for this manuscript focuses on deficiencies concerning B-cell immunity following SFTSV infection in fatal patients, additional data that includes the state of the overall immune system of these patients (T-cells, B-cells, NK cells) could be insightful and further elucidate the mechanisms of SFTSV pathogenesis.

Furthermore, examination of viral load in B-cells could be explaining impaired B-cell-dependent immunity due to virus replication, since there has been a study to report the expression of DC-SIGN, which is known as the cellular receptor for SFTSV entry, on B cells (Rappocciolo et al. 2005 PLoS Pathogens).

Abstract: The authors state in the abstract that SFTSV is an “emerging infectious disease caused by a novel member of phlebovirus.” While this is correct, the sentence should state that it is part of the phlebovirus genera.

Figure presentation: Several of the figures where statistical analysis was performed for the healthy, survived, and deceased groups lack indication of which groups are statistically significant. This can be improved by adding bars indicating which groups are significant. Additionally, the western blot in figure 1 shows the failure of the sera from all patients to react with Gc, which is indeed interesting. A positive control using commercial antibodies or an antibody against the tag epitope of the purified protein could be used as an additional control to ensure equal expression between Gc and Gn.

Response to the reviewers' comments

For reviewer #1

Main criticism:

- 1) In figure 3, only the relative frequencies of the various B cell subsets are presented, but it remains unclear, which subsets show changes in total numbers between the groups. For example, is there really a dramatic loss of naive B cells in the deceased patients, or is the lower frequency only due to higher number of other B cells, with total number of naive B cells actually unchanged? Thus, the total numbers of B cells should also be presented.

Response: To address the reviewer's suggestion, we calculated the data and presented various B cell subsets as the percentage of total B cells as well as the ratio of total B cell in total lymphocytes in Figure 3C-G (the lower panel). The formula for the calculation of the numbers of the various B cell subsets was: the ratio of B cell subset/total B cells multiplied by the ratio of total B cells/total lymphocytes, then multiplied by the amount of the lymphocytes which were provided by the clinical tests at the same day point when the blood samples for flow cytometry analysis were collected. The results between the frequencies and the numbers of the various B cell subsets displayed high consistency as we estimated. Actually, based on the data we previously published, we had found that the count of leukocytes and lymphocytes manifested no significant difference among the patients' groups with different severity in the acute phase of SFTS [1, 2]. We also compared the data of lymphocyte subset analysis of partial patients in clinical detection, and the result demonstrated that the proportion of CD19⁺ B cells in total lymphocytes between survived and deceased group showed no significant differences either (listed in the table below).

Index	Survived (n=13) †	Deceased (n=6) †	p
CD3+/lymphocyte	60.6(27.2-82.0)	55.6(12.0-74.6)	0.284
CD3 ⁺ CD4 ⁺ /lymphocyte	27.1(13.5-41.5)	27.2(14.1-39.5)	0.583
CD3 ⁺ CD8 ⁺ /lymphocyte	29.1(16.5-59.2)	25.3(7.7-37.5)	0.474
CD3 ⁺ CD4 ⁺ /CD3 ⁺ CD8 ⁺	0.90(0.27-1.85)	1.00(0.53-1.44)	0.408
CD19 ⁺ /lymphocyte	20.0(3.6-23.4)	24.0(6.5-34.1)	0.315
CD16 ⁺ CD56 ⁺ /lymphocyte	16.4(7.2-47.3)	14.0(10.1-21.7)	0.424

† lymphocyte subsets of partial patients from clinical detection

- 2) At first glance it is unexpected and may appear even contradictory that the deceased patients have no specific antibody response, but higher frequencies of plasmablasts. This may perhaps be explained by a general B cell activation towards plasmablast differentiation in the absence of a functional specific response. However, as this is a central issue, the plasmablasts need to be better characterized. It is strange that in the deceased patients about 70% and in the surviving patients 30% of plasmablasts are IgM-IgG-. A plasmablast without strong Ig expression is highly questionable. Perhaps, there is an induction of specific class-switching to IgA? There are also so-called IgD-only B cells and plasmablasts/plasma cells in the human that produce IgD in the complete absence of IgM. Can they play a role here? Moreover, the definition of the cells as plasmablasts relies only on the staining pattern CD27-high, CD38-high. Perhaps, in the acute viral infection, some activated B cells transiently upregulate CD27 and CD38 without being plasmablasts? Thus, it is recommended that the CD27⁺⁺CD38⁺⁺ cells are also stained for IgA and IgD, and that the cells are analyzed for key plasmablast/plasma cell markers (upregulation of transcription factors BLIMP1, IRF4, XBP1) by FACS or RT-PCR.

Response: The reviewer's comments are well taken. We conducted the flow cytometry analysis using panel E (CD3, CD19, CD27, CD38, IgA, IgD) , and CD27^{high}CD38^{high} B cells were sorted for gene expression analysis of the key cell markers, including BLIMP1, IRF4 and XBP1 by RT-PCR. The phenotypic analysis

revealed that only low levels of IgD⁺ and IgA⁺ fractions were detected in the PBs and they showed no significant difference between the two patient groups (Figure 4C-4D). Strikingly, the expression of BLIMP1, IRF4 and XBP1 of CD27^{high}CD38^{high} B cells in survived patients were up-modulated simultaneously as compared with their expression in healthy donors. However, all three key transcription factors involved in the maturation of plasmablasts were all dramatically inhibited in deceased patients (Figure 4F and 4G). The results were discussed in the text of the revised manuscript.

3) One of the peripheral blood B cell subsets that showed a significant difference between the two patient groups are CD19⁺CD27⁻IgD⁻ B cells. In healthy individuals, it is well known that these are primarily IgG⁺CD27⁻ B cells, which account for 10-20% of all IgG memory B cells. However, because of the massive B cell compartment alterations present in the patients with acute virus infection, it remains uncertain what the cells defined by lack of CD27 and IgD really are. Are they indeed IgG⁺ memory B cells? Can it be excluded that they are, for example, immature B cells (IgM⁺IgD⁻CD27⁻), released from the bone marrow, or that they are polyclonally activated naive B cells that downregulated IgD upon stimulation? It would likely be sufficient to perform flow cytometric analysis in a few patients and healthy controls to stain the IgD⁻CD27⁻ B cells also for IgG and IgM to clarify their identity better. CD21 could be included to distinguish immature from unusual mature B cells, in case many of them are IgM⁺.

Response: To address the reviewer's questions, we performed analyses as suggested to investigate the compartment of CD19⁺CD27⁻IgD⁻ B cells in the peripheral blood, which showed a significant difference between the two patient groups during acute phase of SFTSV infection. We performed the flow cytometry analysis using panel F (CD19, CD27, IgD, IgM, IgG, CD21), and the results were presented in Figure 3H-3I. We found that the fraction of IgM⁺ cells in the CD27⁻IgD⁻ B cells in the deceased, survived and healthy control group was 26.8%, 19.6% and 31.0% respectively. Meanwhile, the fraction of IgG⁺ cells in the CD27⁻IgD⁻ B cells, representing 4.9%, 3.5% and 1.3% in the respective groups, were significantly lower than the corresponding fraction of IgM⁺ cells. However, both the fraction of IgG⁺

cells and IgM⁺ cells in the CD27⁻IgD⁻ B cells had no significant differences among three groups. Furthermore, according to the reviewer's advice, we measured the CD21 expression in IgM⁺IgD⁻CD27⁻ B cells. Strikingly, the proportion of CD21⁻ cells in IgM⁺IgD⁻CD27⁻ B cells was 89.9% in the deceased patients, significantly higher than 60.9% and 49.3% in the survived and healthy control groups, respectively. Previous study revealed that during B cell ontogeny, bone marrow B cell precursors expressed surface IgM at the early stage, and acquired phenotypical characterization of IgM⁺IgD⁻CD27⁻. In the maturation of naïve B cells, IgM switched to IgD or co-expressed with IgD, leading to naïve B cells displaying two different phenotypes of IgM⁻IgD⁺CD27⁻ or IgM⁺IgD⁺CD27⁻ [3, 4]. Suryani et al have proved that CD21 serves as a good marker which could reflect the extent of maturation of B cells [4]. Therefore, we inferred that IgM⁺IgD⁻CD27⁻ immature B cells would be the major compartment of CD27⁻IgD⁻ B cells. Extremely high frequency of CD21 negative IgM⁺IgD⁻CD27⁻ B cells in the deceased patients might reflect the dysfunction of B cell maturation, and partially explained the significant loss of IgD⁺CD27⁻ naïve B cells during the fatal infection of SFTS virus.

Minor points

a) The complete lack of an anti-humoral immune response against the virus in those patient who died is impressive. Some of the other alterations described may explain or contribute to the failed immune response. However, the authors need to consider that these are so far only correlations. Some of the interpretations need to be tuned down.

Response: We fully agreed with the reviewer's opinion. To illustrate the roles of the virus-specific antibodies, we performed neutralization assays to determine the virus inhibitory activity of sera from recovered patients. In addition, we generated high titer Gn-specific anti-serum in camels immunized with Gn expressed in mammalian cells.. We found that the sera of both convalescent patient and Gn-immunized camel, but not of the deceased patient, potently inhibited SFTS virus infection of Vero cells. This

result provides compelling evidence of the absence of virus-specific antibody response in the pathogenesis of fatal SFTSV infection, as well as the role of Gn-specific antibody in the elimination of this virulent virus. The data were presented in Figure 2C in the revised manuscript.

b) In figure 3B, the horizontal lines above the columns should not extend over the healthy controls, because they are distinct individuals that do not belong the patient groups.

Response: We have corrected this mistake in Figure 3B.

c) On page 10, line 269, it is mentioned that an anti-B220 antibody was used to eliminate B cells from a DC isolation procedure. However, whereas B220 is indeed used as a pan-B cell marker in the mouse, this marker (recognising CD45R) is expressed only by a subset of human B cells. Please explain.

Response: We agreed with the reviewer that B220 expresses only on a subset of human B cells. According to a previous study, there are about 80% human peripheral CD19⁺ B cells expressing B220 [5]. Actually, although a small part of B cells expresses low level of CD11c, its frequency in periphery blood is very small and this small subset of CD11c⁺ B cells is located in T /B cell border in spleen [6]. Another reason we chose B220 was that, besides CD11c⁺MHC II⁺ B cells which need to be excluded from our gating strategy of CD11c⁺ mDCs, recent study demonstrated that thymus-derived $\alpha\beta$ TCR⁺ cells could also express CD11c and MHC class II molecules [7]. Furthermore, earlier evidence has revealed that during acute virus infection, such as Ebola virus, large proportion of T cells were activated [8]. Meanwhile, another study by Bleesing et al found that activated human T cells also up-regulated the expression of B220 [9]. Therefore, using B220 marker, we can exclude most of the B cells and activated T cells. Taken together, we make a brief gating strategy of mDC by B220 marker.

d) In figure 5, necrotic cells are defined as PI⁺ cells, and apoptotic cells as annexin V⁺ cells. However, one typically sees annexin V⁺PI⁻ and annexin V⁺PI⁺ cells. How were the double positive cells grouped? As necrotic or as apoptotic? This needs to be better described.

Response: Considering that propidium iodide (PI) is a fluorescent dye that binds to DNA, early apoptotic cells will exclude PI, while late stage apoptotic cells and necrotic cells will stain PI positively, due to the passage of PI into the nucleus where it binds to DNA. Therefore, we counted the double positive cells as late stage apoptotic cells.

e) At least in the pdf file available for review, the immunofluorescence pictures in figure 5A are very dark and hardly interpretable. The figure should be improved.

Response: We have performed modifications of Figure 5A to improve its visual effect in the revised manuscript.

f) The Discussion is with nearly six pages very long. It would profit from shortening and focusing

Response: The reviewer thinks the discussion is very long and should be shortened and focused. Therefore, we have revised the discussion according to the suggestion.

References:

1. Song, P., et al., *Downregulation of Interferon-beta and Inhibition of TLR3 Expression are associated with Fatal Outcome of Severe Fever with Thrombocytopenia Syndrome*. Sci Rep, 2017. **7**(1): p. 6532.
2. Jia, B., et al., *A scoring model for predicting prognosis of patients with severe fever with thrombocytopenia syndrome*. PLoS Negl Trop Dis, 2017. **11**(9): p. e0005909.
3. Chen, K., et al., *Immunoglobulin D enhances immune surveillance by activating antimicrobial, proinflammatory and B cell-stimulating programs in basophils*. Nat Immunol, 2009. **10**(8): p.

889-98.

4. Suryani, S., et al., *Differential expression of CD21 identifies developmentally and functionally distinct subsets of human transitional B cells*. *Blood*, 2010. **115**(3): p. 519-29.
5. Rodig, S.J., et al., *The CD45 isoform B220 identifies select subsets of human B cells and B-cell lymphoproliferative disorders*. *Hum Pathol*, 2005. **36**(1): p. 51-7.
6. Rubtsov, A.V., et al., *CD11c-Expressing B Cells Are Located at the T Cell/B Cell Border in Spleen and Are Potent APCs*. *J Immunol*, 2015. **195**(1): p. 71-9.
7. Kuka, M., I. Munitic, and J.D. Ashwell, *Identification and characterization of polyclonal alphabeta-T cells with dendritic cell properties*. *Nat Commun*, 2012. **3**: p. 1223.
8. McElroy, A.K., et al., *Human Ebola virus infection results in substantial immune activation*. *Proc Natl Acad Sci U S A*, 2015. **112**(15): p. 4719-24.
9. Bleesing, J.J., et al., *Human T cell activation induces the expression of a novel CD45 isoform that is analogous to murine B220 and is associated with altered O-glycan synthesis and onset of apoptosis*. *Cell Immunol*, 2001. **213**(1): p. 72-81.

For reviewer #2

1. The introduction contains too much irrelevant information and could be shortened.

Response: We have shortened the introduction, and removed some less relevant information.

2. Wrong reference cited for line 104 – 106. That study used non-monocyte derived cell lines for infection.

Response: The correct reference has been used.

3. Revision for Fig 1 and Fig 2:

a. To confirm that antibodies in the survive patients play a role in the clearance of virus and protection, functional neutralization assays with the patients' serum samples should be performed. Otherwise this remains an association and its impact should not be over claimed.

Response: The reviewer's point is well taken. We have performed neutralization analysis to establish the roles of virus-specific serum antibody in inhibiting virus infection. In addition, we also demonstrated Gn-specific antibody generated by immunizing camels with a mammalian expressed Gn protein in inhibin viral infection. The data are presented in Figure 2C in the revised manuscript. Please also see Answer to Reviewer #1, Minor point a.

b. Fig 1, the profiling of NP specific antibody (Ab) is by ELISA and profiling of Gn specific Ab is by western blot. Data of Gn by ELISA is left as unpublished data. To provide better consistency, it would be better to present the ELISA data of Gn together with ELISA data of NP in the main figure and shift the data of Gn western blot to the supplementary with the data from Supp Fig 1.

c. Authors should perform clustering of the data in color chart of the ELISA data. As it stands, the current figure makes it very difficult to read the data.

Response: The reviewer's suggestions are well taken. We moved the Gn Western blot to the supplementary (Fig S1), and instead presented dynamic ELISA data of Gn together with ELISA data of NP in the main figure in the revised manuscript. Meanwhile, we performed clustering of the data in color chart of the ELISA results and generated dynamic profiles of virus-specific antibodies in Figure 1C and 1D in the revised manuscript.

d. Fig 2B, the way the graphs are plotted is not meaningful to observe correlation. Correlation curve plotted in the manner of Fig 2C is sufficient.

Response: We have deleted it according to the reviewer's suggestion.

e. Fig 2C, the number of data points shown do not correspond to the number of data points collected. It is not clear how the authors selected the data to generate the correlation curve. It would be more meaningful to convert all the data points collected in Fig 2B into the correlation plot with just the viral load versus the Ab titre.

Response: That's actually a false appearance caused by the superimposition of many different data points . We have altered the plot to generate a new correlation curve in the current Figure 2B. Detailed information of this new figure was described in the revised manuscript.

4. Revision for Fig 3 and Fig 4:

a. It is possible that the disease could induce leukopenia in patients and diminish the overall B cells numbers in the blood. Authors should provide data of blood counts that includes leucocytes and granulocytes. In addition, the numbers of total B cells should be compared between groups. Without this information, data in Fig 3C is not meaningful, as it is possible that while the % of a specific subset relative to total B cells is the same between survived and deceased groups, the actual numbers could be vastly different due to overall reduction of B cells.

Response: We fully agree with the reviewer's comments.. Based on the data we previously published, although SFTS virus infection could induce leukopenia in nearly all patients, the number of leukocytes and lymphocytes manifested no significant difference among the patients' groups with different severity in the acute phase of SFTS [1, 2]. Additionally, according to the data of lymphocyte subset analysis in clinical samples, we found that the fraction of peripheral CD19⁺ B cells in total lymphocytes also showed no significant difference between survived and deceased group (listed in the table below). In order to make the figure more reflective, we re-calculated the numbers of B cell subsets and presented them in Figure 3C-G (the lower panel) in the revised manuscript. The formula for the calculation of the numbers of the various B cell subsets is: the ratio of B cell subset/total B cells multiplied by the ratio of total B cells/total lymphocytes, then multiplied by the

amount of the lymphocytes which were provided by the clinical tests at the same day point when the blood samples for flow cytometry analysis were collected.

Index	Survived (n=13) †	Deceased (n=6) †	p
CD3+/lymphocyte	60.6(27.2-82.0)	55.6(12.0-74.6)	0.284
CD3 ⁺ CD4 ⁺ /lymphocyte	27.1(13.5-41.5)	27.2(14.1-39.5)	0.583
CD3 ⁺ CD8 ⁺ /lymphocyte	29.1(16.5-59.2)	25.3(7.7-37.5)	0.474
CD3 ⁺ CD4 ⁺ /CD3 ⁺ CD8 ⁺	0.90(0.27-1.85)	1.00(0.53-1.44)	0.408
CD19 ⁺ /lymphocyte	20.0(3.6-23.4)	24.0(6.5-34.1)	0.315
CD16 ⁺ CD56 ⁺ /lymphocyte	16.4(7.2-47.3)	14.0(10.1-21.7)	0.424

† lymphocyte subsets of partial patients from clinical detection

b. The rationale of having 2 different gating strategies for PB and MB was not clearly explained. Authors should consider including the CD38 marker into the analysis presented in Fig 3. In this way, the analysis of B cells subset definition could be standardized.

Response: The reason that two different gating strategies were used was that we applied a two-step approach to illustrate the regulation of B cells during acute phase of SFTS. In the first step (Figure 3), the combination of CD19/CD27/IgD as the B cell marker could distinguish the composition of peripheral B cells, such as naïve, memory and marginal zone-like B cells, as previously reported [3, 4]. We then further defined the plasmablasts as CD19⁺CD27⁺⁺IgD⁻ in the CD19⁺CD27⁺IgD⁻ group, since a previous study by Avery et al reported that increased expression of CD27 on human memory B cell correlated with their commitment to plasma cell lineage [5]. Due to the limitation of our FACS machinery (BD Aira II with six channels), CD38 was left to the second step gating. Therefore, in the second step (Figure 4), we included the CD38 marker in the analysis of plasmablast which are defined as CD27^{high}CD38^{high} B cells.

c. Fig 4, the ELISPOT was done using total B cells. Authors should consider doing the ELISPOT

with sorted PB to demonstrate the lack of virus specific Ab secreting PB cells.

Response: Different from ELISPOT assays that detect cytokines secreted by immune cells, the ELISPOT assay we performed in the present study captures virus-specific IgG stimulation. During the acute phase of SFTS virus infection, plasmablasts are likely the only antibody secreting cell type in patients' peripheral blood. Many previous studies also applied the similar ELISPOT assay in measuring the specific antibody secretion of ASC [6-9].

d. In lines 235 – 239, authors concluded that the observation of increase proportion of IgM-IgG⁻ in the deceased group meant this subset is responsible for abrogated serological response. This is similar to saying that this subset of cells is immunosuppressive, which is not true. In addition, the comparison of this subset was done in % relative to total B cells. Data of % relative to total leucocytes would be more meaningful.

Response: We agree with reviewer's opinion, and the word "responsible" here is actually not accurate. What we intended to say is that both the overwhelming proportion of IgM⁺IgG⁻ PBs would be the results of the humoral response failure in fatal SFTSV infection. We have deleted the confusing sentence in the revised manuscript.

To maintain the consistency of the data presented in the whole article, we directly provided the numbers of CD27⁺CD38⁺ and CD27⁺CD38⁻ subsets in peripheral blood as elsewhere in this manuscript.

5. Revision for Fig 5:

a. Author presented 2 effects to the mDCs: apoptosis/necrosis and the suppression of co-stimulatory molecules that could suppress Ag presentation. The manuscript would flow better if the text from lines 258 – 283 is separated into 2 sub-sections based on these 2 effects.

Response: The reviewer's suggestion is well taken, and we have separated the text

into 2 sub-sections based on these 2 effects in the revised manuscript.

b. Regarding the apoptosis assay, authors allowed the PBMCs to adhere and subsequently performed staining on all adherent cells with the assumption they are monocytes. This is incorrect. The authors would need to either proof that this method produces homogenous monocytes in the adherent cells or add in markers to confirm the identity of the cells undergoing apoptosis. A more direct alternative method would be to track apoptosis using a annexin V kit by flow cytometry using the same gating strategy used for the mDCs.

Response: According to previous studies, the adherent cells produced by this method usually contain about 90% of monocytes[10, 11], they might include a small part of dendritic cells and macrophage, even as well as a very minor fraction of CD3⁺ cells (<3%) [10]. We have changed the term “monocytes” into “adherent cells of PBMC”. Considering the present study demonstrated that the apoptosis and necrosis rates of the adherent cells in the deceased patients were significantly higher than 10%, representing 32.6% and 18.3%, respectively, we could still infer that the monocytes in the deceased patients undergo the significant apoptosis and necrosis, as compared with in that the survived patients.

The reviewer’s suggestion on using a annexin V kit by flow cytometry is an excellent suggestion. Unfortunately the patient PBMCs are cryogenically preserved in small quantity and the freeze-thawing has detrimental effects on APC and could induce apoptosis and necrosis of dendritic cells as previously described [12, 13]. We are unable to carry out the experiments suggested by the reviewer. Instead, we isolated sufficient mDCs (purify>97%) from fresh peripheral blood of healthy donors using CD1c (BDCA-1) Dendritic Cell Isolation Kit (Miltenyi Biotec, Germany), and carried out *in vitro* infection experiments to analyze apoptosis and performed function assays on mDC. The results of apoptosis of mDC using a annexin V kit by flow cytometry are presented in Figure 6C-6D in the revised manuscript. Details of the experiment are provided in the revised manuscript.

C. In the second part, authors profiles the co-stimulatory molecules and proposed that the Ag presentation of mDCs is impaired. This is inadequate and more direct function assays would be required to drive this point. Particularly, authors should sort out mDCs from the 2 groups of patients, and assess their ability to present Ag to T cells either by CFSE tagging to the T cells or by ELISOT assay.

Response: The reviewer's point is well taken. As mentioned in the last paragraph, there were some difficulties in performing functional assay using the cryo-preserved patients' samples. Therefore, we performed the allogenic DC/T-cell coculture and stimulation assay *in vitro* using the peripheral blood of healthy donors to investigate the impaired Ag presentation of mDCs caused by SFTS virus infection as previously described [15-17]. The proliferation rate of CFSE-tagged CD4⁺ T cells and the level of IFN- γ in the supernatant after allogenic DC stimulation are determined and presented in Figure 6I-6J. The results provided compelling evidence that Ag presentation function of mDCs was impaired due to SFTS virus infection.

d. Lastly, an important fundamental concept must be addressed. Myeloid DC (also known as conventional DC) is derived from CDP and not monocytes.

Response: We fully agree with the reviewer's comment. Although monocyte derived DC (mo-DC) share some common phenotypic markers with mDC, including CD11c and MHC class II molecules, they actually originate from different precursor cells [18]. To avoid conceptual confusion between mDC and mo-DC, we have deleted the incorrect expression and corrected the sentence "Severe apoptosis of monocytes followed by the failure of myeloid DC differentiation" in the revised text.

6. Fig 6, authors proposed that B cells Ag presentation to Tfh is vital for Tfh expansion and the alteration of CD80 and HLA-DR disrupted this process. This unique process happens exclusively in the lymph nodes and spleen in a small percentage of B cells within the interfollicular zone². It is questionable if the profiling of co-stimulatory molecules in the peripheral B cells would be

sufficient to drive this conclusion. Direct immunofluorescent staining of B cells and Tfh co-localization in the interfollicular zone would be required to show the abrogation of this interaction.

Response: The reviewer's points are well taken. However, since most SFTS patients suffered from dysfunction of coagulation to varying extents [1], the biopsy of LN or SP would likely run the risk of hemorrhage on SFTS patients, especially on severe individuals. In addition, patients died of infectious diseases are rarely allowed for biopsy. Therefore, we haven't been able to do direct immunofluorescent staining of B cells and Tfh co-localization assay on patient's lymph nodes. Previous study demonstrated that antigen-presenting human B cells could also exist in the peripheral blood, and expand in inflammatory conditions [21]. Thus, as an indirect evidence, we measured the expression of CD80 and HLA-DR in peripheral B cell subsets to assess the ability of B cell antigen presentation during SFTS virus infection. After all, because the transcriptional and phenotypic similarities between peripheral and follicular B cells involved in antigen presenting process are still poorly understood, the profiling of co-stimulatory molecules in the peripheral B cells would only serve as indirect evidence for this process. Some of the interpretations have been revised to address the reviewer's concerns in the revised manuscript.

7. Fig 7, where authors argued the loss of function Tfh in deceased patients would need to be further addressed:

a. Representative pseudoplot provided highlighted only 2 IL-21 positive staining events. This is too low to be conclusive if we consider the noise to actual signal ratio of a generic flow cytometry machine. In view of this, authors should consider using IL-21 gene expression from sorted pTfh cells.

Response: Actually, although the number of IL-21 positive staining events of this pseudoplot which represents one deceased individual is significantly lower than the number of some survived individuals, there are more than 2 events in it and some of

the events are neglected due to their plots are too minor to read. We have enhanced the visual effect of the pseudoplot by lightening all the IL-21 positive staining events, and presented it in Figure 5A in the revised manuscript. To further address reviewer's concern, we have sorted pTfh cells of SFTS patients, and measured IL-21 gene expression by RT-PCR. The result presented in Figure 5D (right panel) in the revised manuscript demonstrates that the level of IL-21 mRNA in pTfh cells of survived patients is significantly higher than the level of deceased patients. This result further supports the conclusion that the function of pTfh cells is impaired during fatal SFTS virus infection.

b. Following the reference cited by the authors, functional role of pTfH in patients was shown by their ability to induce phenotypic changes in naïve B cells through co-culture system. They should consider using a similar system to assign a function difference to the pTfH beyond the profiling of IL21.

Response: Taking the reviewer's suggestion, we sorted naïve B cells and pTfH cells from peripheral blood of patients by flow cytometry. pTfH cells were cultured in RPMI-1640 supplemented with 10% heat-inactivated FBS and stimulated with 1ul Leukocyte Activation Cocktail at 37°C for 3h. CFSE was added to naïve B cell culture at a final concentration of 500nM and incubated at 37°C for 7 min. After being washed twice, stimulated pTfH cells were co-cultured with autologous CFSE-tagged naïve B cells tagged at 1:1 ratio at 37°C for 7 days. The proliferation rate of, CD38 and IgM expression on the B cells were measured by flow cytometry. Different from the results that we cited in the manuscript, we only observed the significant difference of B cell proliferation rate between the survived and deceased patients. The results were presented in Figure 5E-5F in the revised manuscript.

8. Lastly, it would make better sense to change the order of some of the data. Particularly, pTfH data should come after showing Ab suppression, followed by the B cell Ag presentation and mDC data which explain the change in Tfh.

Response: We appreciate the suggestions and did the suggested re-arrangement in the revised manuscript.

References:

1. Song, P., et al., *Downregulation of Interferon-beta and Inhibition of TLR3 Expression are associated with Fatal Outcome of Severe Fever with Thrombocytopenia Syndrome*. *Sci Rep*, 2017. **7**(1): p. 6532.
2. Jia, B., et al., *A scoring model for predicting prognosis of patients with severe fever with thrombocytopenia syndrome*. *PLoS Negl Trop Dis*, 2017. **11**(9): p. e0005909.
3. Joo, H., et al., *Serum from patients with SLE instructs monocytes to promote IgG and IgA plasmablast differentiation*. *J Exp Med*, 2012. **209**(7): p. 1335-48.
4. Budeus, B., et al., *Complexity of the human memory B-cell compartment is determined by the versatility of clonal diversification in germinal centers*. *Proc Natl Acad Sci U S A*, 2015. **112**(38): p. E5281-9.
5. Avery, D.T., et al., *Increased expression of CD27 on activated human memory B cells correlates with their commitment to the plasma cell lineage*. *J Immunol*, 2005. **174**(7): p. 4034-42.
6. McElroy, A.K., et al., *Human Ebola virus infection results in substantial immune activation*. *Proc Natl Acad Sci U S A*, 2015. **112**(15): p. 4719-24.
7. Nahrendorf, W., et al., *Memory B-cell and antibody responses induced by Plasmodium falciparum sporozoite immunization*. *J Infect Dis*, 2014. **210**(12): p. 1981-90.
8. Muema, D.M., et al., *Control of Viremia Enables Acquisition of Resting Memory B Cells with Age and Normalization of Activated B Cell Phenotypes in HIV-Infected Children*. *J Immunol*, 2015. **195**(3): p. 1082-91.
9. Sasaki, S., et al., *Limited efficacy of inactivated influenza vaccine in elderly individuals is associated with decreased production of vaccine-specific antibodies*. *J Clin Invest*, 2011. **121**(8): p. 3109-19.
10. Pappasavvas, E., et al., *IL-13 acutely augments HIV-specific and recall responses from HIV-1-infected subjects in vitro by modulating monocytes*. *J Immunol*, 2005. **175**(8): p. 5532-40.

11. Laval, K., et al., *Equine Herpesvirus Type 1 Enhances Viral Replication in CD172a+ Monocytic Cells upon Adhesion to Endothelial Cells*. J Virol, 2015. **89**(21): p. 10912-23.
12. Nile, C.J., et al., *Expression and regulation of interleukin-33 in human monocytes*. Immunology, 2010. **130**(2): p. 172-80.
13. Owen, R.E., et al., *Loss of T cell responses following long-term cryopreservation*. J Immunol Methods, 2007. **326**(1-2): p. 93-115.
14. Jia, B., et al., *A scoring model for predicting prognosis of patients with severe fever with thrombocytopenia syndrome*. PLoS Negl Trop Dis, 2017. **11**(9): p. e0005909.
15. Azzaoui, I., et al., *CCL18 differentiates dendritic cells in tolerogenic cells able to prime regulatory T cells in healthy subjects*. Blood, 2011. **118**(13): p. 3549-58.
16. Stocki, P., X.N. Wang, and A.M. Dickinson, *Inducible heat shock protein 70 reduces T cell responses and stimulatory capacity of monocyte-derived dendritic cells*. J Biol Chem, 2012. **287**(15): p. 12387-94.
17. Gonzalez, P.A., et al., *Respiratory syncytial virus impairs T cell activation by preventing synapse assembly with dendritic cells*. Proc Natl Acad Sci U S A, 2008. **105**(39): p. 14999-5004.
18. Helft, J., et al., *GM-CSF Mouse Bone Marrow Cultures Comprise a Heterogeneous Population of CD11c(+)MHCII(+) Macrophages and Dendritic Cells*. Immunity, 2015. **42**(6): p. 1197-211.
19. Barnett, L.G., et al., *B cell antigen presentation in the initiation of follicular helper T cell and germinal center differentiation*. J Immunol, 2014. **192**(8): p. 3607-17.
20. Rubtsov, A.V., et al., *CD11c-Expressing B Cells Are Located at the T Cell/B Cell Border in Spleen and Are Potent APCs*. J Immunol, 2015. **195**(1): p. 71-9.
21. Shimabukuro-Vornhagen, A., et al., *Antigen-presenting human B cells are expanded in inflammatory conditions*. J Leukoc Biol, 2017. **101**(2): p. 577-587.

For reviewer #3

Minor Comments:

The rationale for this manuscript is well founded based on a previous study by the same authors in 2017, where they found the reduction of both IFN- β and IL-1 β in patients that have a severe or fatal outcome of the disease. Furthermore, they observed a reduction of myeloid dendritic cells

and an overall suppression of TLR3 in the primary targets for SFTSV, myeloid dendritic cells, which are primary targets for SFTSV infection, and an overall suppression of TLR3 in DCs. Hence, the authors postulated that antigen presentation was inhibited in patients that had a fatal outcome following SFTSV infection. While the premise for this manuscript focuses on deficiencies concerning B-cell immunity following SFTSV infection in fatal patients, additional data that includes the state of the overall immune system of these patients (T-cells, B-cells, NK cells) could be insightful and further elucidate the mechanisms of SFTSV pathogenesis.

Furthermore, examination of viral load in B-cells could be explaining impaired B-cell-dependent immunity due to virus replication, since there has been a study to report the expression of DC-SIGN, which is known as the cellular receptor for SFTSV entry, on B cells (Rappocciolo et al. 2005 PLoS Pathogens).

Abstract: The authors state in the abstract that SFTSV is an “emerging infectious disease caused by a novel member of phlebovirus.” While this is correct, the sentence should state that it is part of the phlebovirus genera.

Figure presentation: Several of the figures where statistical analysis was performed for the healthy, survived, and deceased groups lack indication of which groups are statistically significant. This can be improved by adding bars indicating which groups are significant. Additionally, the western blot in figure 1 shows the failure of the sera from all patients to react with Gc, which is indeed interesting. A positive control using commercial antibodies or an antibody against the tag epitope of the purified protein could be used as an additional control to ensure equal expression between Gc and Gn.

Responses:

1. To address the reviewer’s questions on the state of the overall immune system of SFTS patients, such as T-cells, B-cells and NK cells, we analyzed the subsets of lymphocytes from SFTS patients by flow cytometry, and found that the main subsets of peripheral lymphocytes, including total T cells, total B cells and NK cells,

manifested no significant differences among patients' groups regardless of the disease severity (presented in the table below). Differences were observed only for specific cell subsets, such as PBs and Tfh.

Index	Survived (n=13) †	Deceased (n=6) †	p
CD3+/lymphocyte	60.6(27.2-82.0)	55.6(12.0-74.6)	0.284
CD3 ⁺ CD4 ⁺ /lymphocyte	27.1(13.5-41.5)	27.2(14.1-39.5)	0.583
CD3 ⁺ CD8 ⁺ /lymphocyte	29.1(16.5-59.2)	25.3(7.7-37.5)	0.474
CD3 ⁺ CD4 ⁺ /CD3 ⁺ CD8 ⁺	0.90(0.27-1.85)	1.00(0.53-1.44)	0.408
CD19 ⁺ /lymphocyte	20.0(3.6-23.4)	24.0(6.5-34.1)	0.315
CD16 ⁺ CD56 ⁺ /lymphocyte	16.4(7.2-47.3)	14.0(10.1-21.7)	0.424

† lymphocyte subsets of partial patients from clinical detection

2. According to a previous study, SFTSV could use DC-SIGN as cellular receptor for entry [1]. Meanwhile, DC-SIGN has been proved to be expressed on B cells [2]. To investigate if SFTSV infects B cells, we investigated the sorted B cells of both survived (n=5) and deceased patients (n=4) by RT-PCR for viral RNA, but we failed to detect the viral RNA in all samples. In addition, we isolated CD19⁺ B cells from the peripheral blood of healthy donors by magnetic beads and performed *in vitro* infection assay. Consistent with the result of patients' samples, we also failed to detect the viral RNA in the B cells infected *in vitro*. Therefore, we do not have evidence that SFTS virus could productively infect B cells.

3. The reviewer's suggestion on phlebovirus is well taken and we have revised "phlebovirus" to "phlebovirus genera" following the reviewer's suggestion in the revised manuscript.

4. The reviewer's comments are well taken. We have provided explanation in the relevant figure legends to indicate statistic significance.

Actually, in our previous work, Gn and Gc has been identified by Western Blot analysis, and anti-His antibody (Santa Cruze, USA) served as the primary antibody at a dilution of 1:300 (Figure S1A and S1B). The result has been presented in Figure S1A and S1B in the revised manuscript.

References:

1. Hofmann, H., et al., *Severe fever with thrombocytopenia virus glycoproteins are targeted by neutralizing antibodies and can use DC-SIGN as a receptor for pH-dependent entry into human and animal cell lines*. J Virol, 2013. **87**(8): p. 4384-94.
2. Rappocciolo, G., et al., *DC-SIGN on B lymphocytes is required for transmission of HIV-1 to T lymphocytes*. PLoS Pathog, 2006. **2**(7): p. e70.

REVIEWERS' COMMENTS:

Reviewer #1 (Remarks to the Author):

The point raised in my report for the first version of the manuscript have been adequately addressed. The addition stainings of B cell subsets make the distorted B cell compartment in the deceased patients more clear.

Reviewer #2 (Remarks to the Author):

General comments:

The authors have made efforts to address most of the previous concerns with new in vitro experiments. This reviewer is aware of the technical limitations in working with cryopreserved patient's samples. However, the use of peripheral B cells as an indirect evidence to explain antigen presentation to T_H, a process that happens exclusively in the secondary lymphoid organs, is overarching. Usually, organ specific studies require the use of animal models for such conclusions. There remains a few minor points related to data presentation.

Specific comments:

1. Figure 2B - the way the x-axis is presented is confusing. Particularly, it gives an impression that the lower the antibody titer, the lower the viral load, which is opposite to what the authors have written. I believe those x-axis values are the highest dilution factor of the patient serum that gives detectable signal, i.e the lower the values the more antibody the patients have. If this is the case, please address this properly and mention it in the figure legends.
2. Figure 3 - please provide a legend to indicate what is black, red and blue line in panel 3c.
3. Figure 6c and d - instead of measuring total annexin v and PI percentage, the proper way to analyze the data would be to quantify early apoptotic cells (Annexin V+/PI negative) and late apoptotic cells (Annexin V+/ PI+).
4. Figure 6e - the pseudoplot showing HLADR to CD14 & B220 has some compensation issues which could be easily corrected by the analysis software.

Reviewer #3 (Remarks to the Author):

The authors have done a thorough job of revising the present manuscript and their replies to the reviewer's comments are reasonable.

Answers to Reviewer 2#

General comments:

The authors have made efforts to address most of the previous concerns with new in vitro experiments. This reviewer is aware of the technical limitations in working with cryopreserved patient's samples. However, the use of peripheral B cells as an indirect evidence to explain antigen presentation to T_{fh}, a process that happens exclusively in the secondary lymphoid organs, is overarching. Usually, organ specific studies require the use of animal models for such conclusions. There remains a few minor points related to data presentation.

Response: We fully agree with the reviewer's opinion that B cell antigen presentation to T_{fh} is a process happening exclusively in the secondary lymphoid organs. However, due to the surgical risk in the hemorrhagic patients, such an experiment is not practical and better investigated in an animal model. Therefore, to address the reviewer's concern, we revised some interpretations in the Results and the Discussion to reflect the limitations of the study in the revised manuscript.

Specific comments:

1. Figure 2B - the way the x-axis is presented is confusing. Particularly, it gives an impression that the lower the antibody titer, the lower the viral load, which is opposite to what the authors have written. I believe those x-axis values are the highest dilution factor of the patient serum that gives detectable signal, i.e. the lower the values the more antibody the patients have. If this is the case, please address this properly and mention it in the figure legends.

Response: The reviewer's suggestion is well taken. We have changed the way of the presentation of x-axis in the revised manuscript.

2. Figure 3 - please provide a legend to indicate what is black, red and blue line in

panel 3c.

Response: The original description of the colored lines for Fig. 3c was in the figure legend (lines 1021-1023, revised manuscript).

3. Figure 6c and d - instead of measuring total annexin v and PI percentage, the proper way to analyze the data would be to quantify early apoptotic cells (Annexin V⁺/PI⁻ negative) and late apoptotic cells (Annexin V⁺/PI⁺).

Response: The reviewer's point is well taken. We have measured the proportions of Annexin V⁺/PI⁻ and Annexin V⁺/PI⁺ cells in total mDCs, as suggested by the reviewer and presented them in the revised Figure 6C. We have also altered related expressions in the Results and Figure legend in the revised manuscript.

4. Figure 6e - the pseudoplot showing HLADR to CD14 & B220 has some compensation issues which could be easily corrected by the analysis software.

Response: We have corrected the compensation issues in the pseudoplot showing HLA-DR to CD14 & B220 (Figure 6E) by the analysis software of the flow cytometer.